# Evaluating normative representation learning in generative AI for robust anomaly detection in brain imaging

Cosmin I. Bercea [1,2] ✉, Benedikt Wiestler [3,4], Daniel Rueckert[4,5,6] & Julia A. Schnabel [1,2,4,7]

Normative representation learning focuses on understanding the typical anatomical distributions from large datasets of medical scans from healthy individuals. Generative Artificial Intelligence (AI) leverages this attribute to synthesize images that accurately reflect these normative patterns. This capability enables the AI allowing them to effectively detect and correct anomalies in new, unseen pathological data without the need for expert labeling. Traditional anomaly detection methods often evaluate the anomaly detection performance, overlooking the crucial role of normative learning. In our analysis, we introduce novel metrics, specifically designed to evaluate this facet in AI models. We apply these metrics across various generative AI frameworks, including advanced diffusion models, and rigorously test them against complex and diverse brain pathologies. In addition, we conduct a large multi-reader study to compare these metrics to experts' evaluations. Our analysis demonstrates that models proficient in normative learning exhibit exceptional versatility, adeptly detecting a wide range of unseen medical conditions. Our code is available at https://github.com/compai-lab/2024-ncomms-bercea.git.

The continuous advancement in medical imaging technology has markedly enhanced our ability to diagnose diseases. Yet, this progress presents a new challenge: extracting actionable insights from the vast volumes of medical imaging data now available[1]. This scenario underscores the urgency for automated diagnostic tools capable of efficiently processing this data to offer accurate and timely diagnoses, thereby easing the burden on healthcare systems. Computer-aided diagnostics, particularly those employing supervised learning[2,3], have represented a significant leap in this direction, enabling machines to recognize disease patterns across various imaging modalities[4]. However, these methods often struggle to fully capture the complexity and rarity of human pathologies (https://rarediseases.info.nih.gov/about), especially in the absence of large, annotated datasets[5].

In response to these limitations, Unsupervised Anomaly Detection (UAD)[6] has gained prominence, offering the promise of autonomous anomaly detection without reliance on labeled data. The potential of UAD extends across diverse imaging modalities, from brain MRIs[7–10] to chest X-rays[11,12] and beyond[13–15], suggesting a transformative change in diagnostic approaches. Nonetheless, the clinical integration of UAD faces challenges, including biases toward certain pathology profiles[16] and the opaque nature of these 'black box' systems.

Generative AI[17] has brought a novel dimension to anomaly detection by adeptly capturing the nuances of what is considered 'normal' in medical images, see Fig. 1. The true innovation of generative AI lies in normative representation learning, a concept driven by data to uncover characteristics of a healthy population. Here, anomaly

[1]Chair of Computational Imaging and AI in Medicine, Technical University of Munich (TUM), Munich, Germany. [2]Helmholtz AI and Helmholtz Center Munich, Munich, Germany. [3]Chair of AI for Image-Guided Diagnosis and Therapy, TUM School of Medicine and Health, Munich, Germany. [4]Munich Center for Machine Learning (MCML), Munich, Germany. [5]Chair of AI in Healthcare and Medicine, Technical University of Munich (TUM) and TUM University Hospital, Munich, Germany. [6]Department of Computing, Imperial College London, London, UK. [7]School of Biomedical Engineering and Imaging Sciences, King's College London, London, UK. ✉e-mail: cosmin.bercea@tum.de

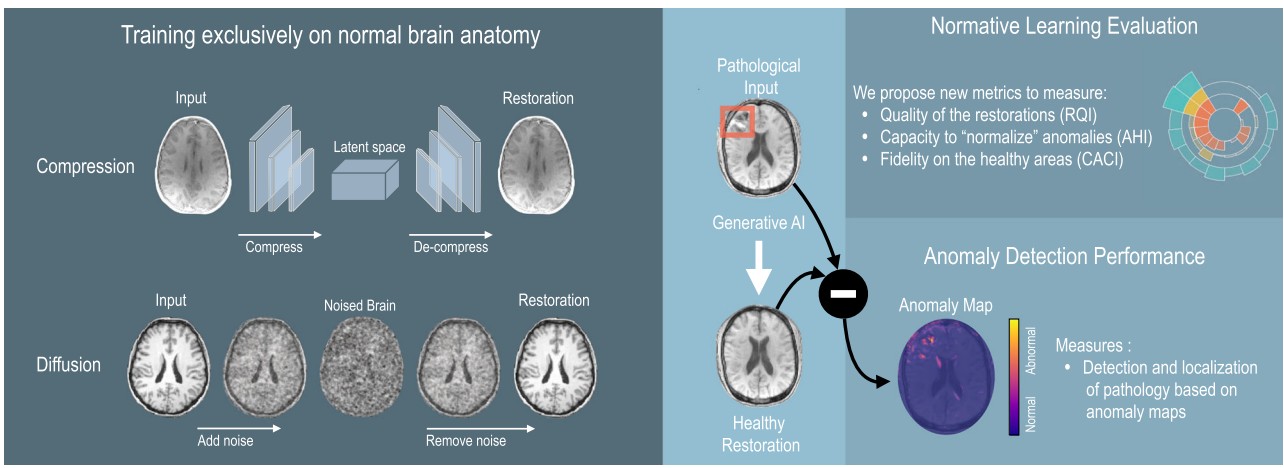

a). Training on healthy data      b). Inference on pathological data

**Fig. 1 | Generative AI for unsupervised anomaly detection in medical imaging. a** demonstrates the AI model being trained on a dataset of normal brain anatomy, a process known as normative representation learning. This can be achieved using compression, e.g., autoencoders, or diffusion processes. In Fig. **b**, a pathological

brain magnetic resonance image (MRI) is input into the AI system, which then outputs a pseudo-healthy restoration of the brain. The transition from pathological input to healthy output is evaluated to produce an anomaly map, highlighting the areas of deviation.

detection assumes a complementary role, focusing on pinpointing deviations from these learned representations. However, the evaluation of generative AI methods often tilts disproportionately towards anomaly detection, leading to biased or incomplete assessments. For instance, certain approaches optimize solely for hyper-intense lesion detection[18], anchor their design principles (e.g., specific noise types) around large tumors[19], or approximate the unknown anomaly distribution through self- or weak-supervision[20,21]. While such methodologies can be advantageous for detecting specific pathologies, they often fall short in broader anomaly detection contexts[16,22].

Addressing this critical gap, our analysis advocates for a reorientation in the evaluation of generative AI, emphasizing its intrinsic role in normative representation learning. This approach marks a shift in the assessment of generative AI, positioning it not just as a tool for disease identification but as a system for evaluating the realness and plausibility of generated counterfactuals in medical imaging.

## Proposed metrics

We propose specific metrics to assess the quality of the pseudo-healthy restorations: Restoration Quality Index (RQI): is an image-based metric that evaluates the perceived quality of synthesized images by measuring their semantic similarity to original inputs.

Anomaly to Healthy Index (AHI): measures the closeness of the distribution of restored pathological images to a healthy reference set.

Healthy Conservation and Anomaly Correction Index (CACI): measures the effectiveness of models in maintaining integrity in healthy regions and correcting anomalies in pathological areas.

These metrics collectively provide a more comprehensive evaluation, extending beyond simple anomaly mapping to include assessments of the quality and accuracy of the normative and pseudo-healthy representations generated by AI models.

## Evolution of generative AI for UAD

The impact of generative AI on medical imaging for anomaly detection has unfolded in distinct evolutionary phases, each marking a pivotal advancement in the field, as illustrated in Fig. 2.

Autoencoders (AEs) set the basis for UAD, premised on the hypothesis that anomalies would induce higher reconstruction errors[4,23–26]. Despite their promise, AEs struggle with generalizing beyond training data without losing detail, leading to anomaly misidentification[26–28].

Variational AEs (VAEs) introduced probabilistic constraints on latent variables, advancing anomaly detection through nuanced posterior distribution approximations[8,28–30]. However, challenges in high-dimensional data representation still impact reconstruction quality.

Generative Adversarial Networks (GANs) revolutionized data synthesis through adversarial training, significantly enhancing anomaly detection[31–34]. However, their tendency towards mode collapse and limitations in preserving healthy tissue are critical areas for improvement.

Hybrid Models, such as Adversarial AEs (AAEs), combine structured latent spaces of VAEs with the superior image generation of GANs[16,35–37]. However, they still face challenges in handling discrepancies in healthy regions.

Diffusion Models have revamped generative modeling by capturing complex characteristics without latent space constraints[38]. Within medical anomaly detection[19,21], diffusion models incrementally add noise to a pathological input, obscuring anomalies up to a certain threshold before methodically reverting them to a pseudo-healthy state. However, the choice of the noise level remains an important challenge[39,40].

Guided Restoration Techniques utilize context-encoding and masking strategies to enhance the accuracy of diagnostics. Incorporating shape priors derived from healthy structures[27] and techniques like random masking in masked AEs (MAEs) and Patched Diffusion models (pDDPM)[41–44] offer nuanced adaptability in anomaly detection. Recent advancements include automatic masking strategies that intelligently transform only regions likely to contain anomalies, preserving the integrity of healthy tissue[40,45].

## Results

In our evaluation of generative AI models, we utilized normal T1-weighted brain MRI datasets, FastMRI+[46] with 176 scans and 581 samples from IXI, for model training. For the evaluation phase, we focused on two key datasets: the enhanced FastMRI+ dataset, which encompasses a wide spectrum of 171 brain pathologies, and 420 subjects from the ATLAS v2.0 dataset[47], known for its diverse range of stroke lesions. This strategy allowed us to rigorously test the capabilities of different models in detecting and localizing a broad array of anomalies, benchmarking their performance against the complexity and diversity of real-world brain pathologies. See Supplementary Figs. 1 and 2 for more details.

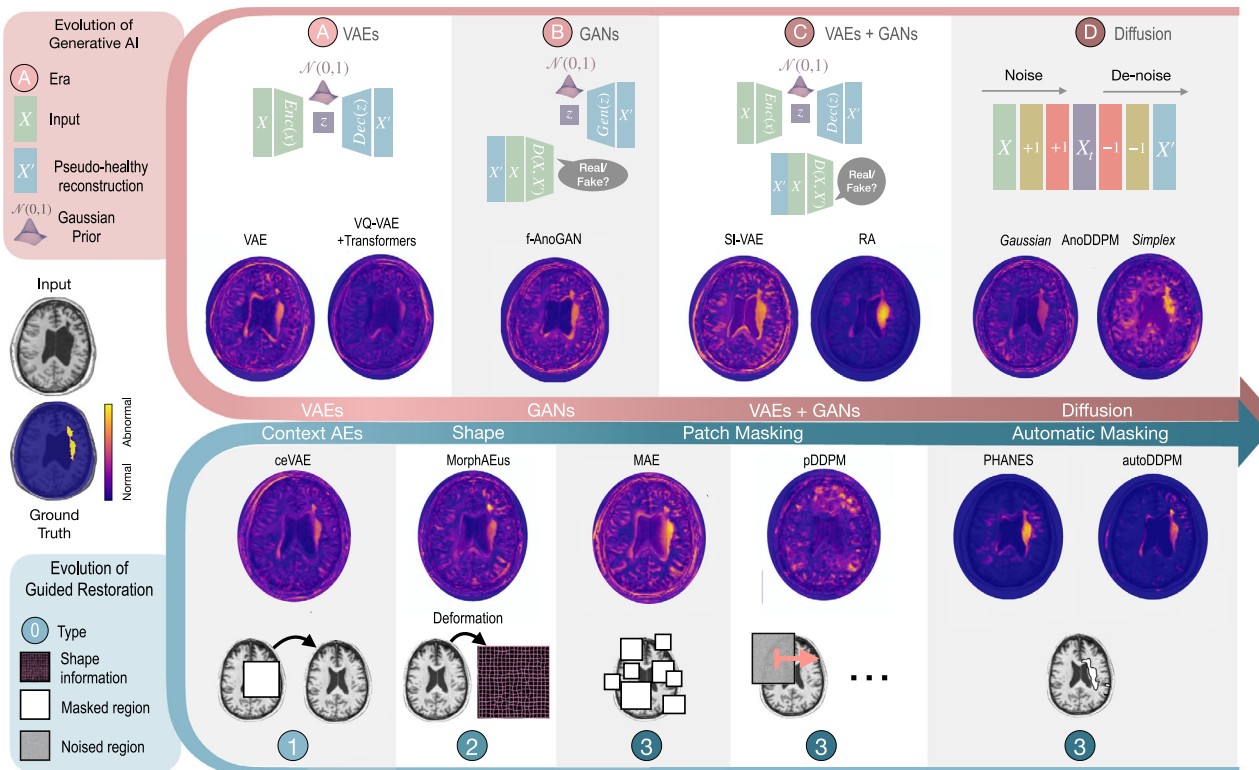

**Fig. 2 | Advancements in generative AI for medical anomaly detection: a chronological perspective.** The timeline showcases the evolution of generative AI techniques in medical imaging, classified into distinct `Eras' from Variational Autoencoders (VAEs) to Diffusion models. The lower segment details the evolution of guided restoration, highlighting the shift from basic deformations to sophisticated automatic masking strategies. Corresponding anomaly maps for each category are provided, allowing direct visual comparison of their detection capabilities in identifying stroke lesions.

## Normative learning evaluation

Our analysis primarily centered on normative representation learning, a crucial aspect for ensuring that models accurately represent healthy anatomical patterns. Figure 3 summarizes the results.

Each of these indices offers a unique perspective on the performance of the generative models. For instance, methods that simply replicate input images, like AEs, achieve high RQI but score low on AHI and CACI. Conversely, methods such as VAEs, latent transformer models (LTMs), and MAEs that remove anomalies and improve CACI often produce blurry outputs, resulting in poor RQI. The AHI metric, which requires the synthesized image distributions to closely resemble a healthy set, proves challenging for many methods, often resulting in near-zero scores. Notably, the FID is particularly demanding; even methods that produce realistic healthy images but lack diversity or show slight domain shifts, such as RA or DDPM-G, find it challenging to achieve good scores. Guided restoration techniques using intelligent masking tend to achieve the best scores.

Therefore, it is crucial to consider these metrics collectively rather than in isolation. Optimal performance is characterized by high scores across RQI, AHI, and CACI, indicating the comprehensive ability to understand and replicate healthy anatomical structures while effectively identifying and rectifying anomalies. To fuse the metrics, we propose a harmonic mean between RQI and CACI, averaged with AHI. This approach balances image quality and anomaly correction while mitigating the impact of near-zero AHI scores on the overall evaluation (see Eq. (4)).

## Anomaly detection performance

The anomaly detection results, detailed in Table 1, reveal notable insights into the performance of various generative AI models (please refer to Supplementary Table 1 for the complete results). PHANES and AutoDDPM, in particular, demonstrated exceptional proficiency in mastering normative aspects of medical imaging, which translated effectively into their leading roles in anomaly detection. In the FastMRI + dataset, AutoDDPM achieved great success, detecting 159 out of 171 pathologies. PHANES showed superior performance in identifying enlarged ventricles and was especially effective in segmenting large stroke lesions in the ATLAS dataset. In contrast, AutoDDPM exhibited heightened sensitivity to smaller stroke lesions. These findings highlight the predictive power of high normative learning scores, as measured by RQI, AHI, and CACI, in determining the capability of a model to navigate the complexities of medical image analysis.

Interestingly, the results further reveal the varying effectiveness of models with average or lower scores in normative learning metrics. These models exhibited inconsistent results across different datasets, suggesting a dependency on the specific pathology types and dataset characteristics. For instance, the MAE model, despite ranking 7th in stroke lesion detection and showing commendable performance in identifying large lesions, ranked only 12th in the FastMRI+ benchmark for detecting more diverse and subtle anomalies like edema or smaller lesions. This inconsistency in performance across varying conditions underscores the critical importance of considering RQI, AHI, and CACI scores collectively to assess the generalization ability in anomaly detection across a wide spectrum of pathologies.

## Interplay between normative learning and anomaly detection

Our in-depth analysis, detailed in Fig. 4, delves into the relationship between normative learning metrics and anomaly detection metrics (⌈*Dice*⌉ or F1 as in Table 1), revealing key insights:

The analysis, visually represented in the chord diagram and top heatmap of Fig. 4 underscores the crucial role of normative learning metrics in universal anomaly detection. Notably, while individual metrics like RQI provide valuable insights, their diagnostic impact is limited when considered in isolation. For instance, models focusing

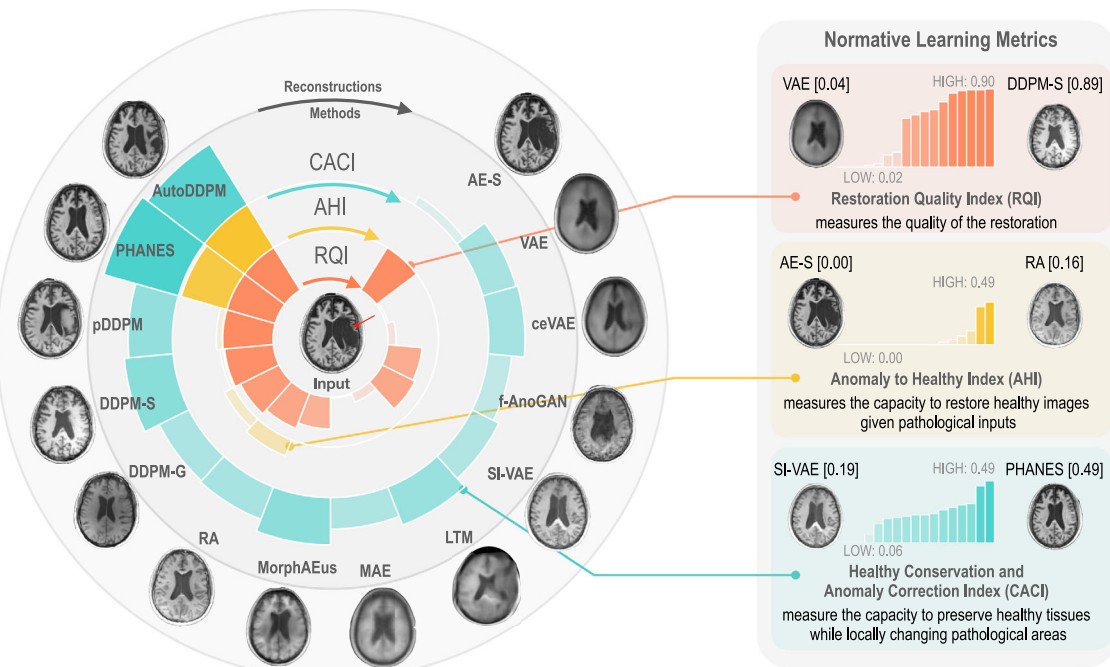

**Fig. 3 | Normative learning evaluation.** The central pathological MRI image is encircled by the pseudo-healthy restorations of different models, with their respective dataset-wide metric scores (RQI, AHI, CACI) radiating outward. Segments with darker and higher values indicate better performance, highlighting effectiveness in image restoration, anomaly normalization, and preservation of healthy tissue. Source Data are provided as a Source Data File.

solely on RQI might replicate anomalies, and missing critical detections. Conversely, CACI shows a high predictive value for anomaly detection. However, the most comprehensive insights are obtained when these metrics are collectively analyzed, with combinations such as CACI and RQI, and particularly the integration of all three metrics (RQI, AHI, CACI), demonstrating enhanced predictive anomaly detection power. This highlights the importance of a balanced normative learning approach in generalizing across various pathologies. Interestingly, we observed a single negative correlation where higher normative metrics were associated with reduced detection of cases labeled as 'paranasal sinus opacification'. Upon review by an expert neuroradiologist, these cases exhibited no clear visual signs of pathology. Methods with a tendency for more false positives might mistakenly detect issues in these cases, likely attributable to the proximity of the affected area to the skull-a region prone to false positives.

Our analysis of the five leading methods (Top 5), as per normative learning metrics, led to two significant observations. First, we noticed negative correlations in the detection of white matter lesions (WML). Expert neuroradiologists pointed out that some lesions were not visible, often obscured by significant motion artifacts, which are not typically classified as anomalies. Second, a higher AHI seemed to negatively affect the quantification of extensive stroke lesions. This trend may be related to how these lesions are annotated and evaluated. Pathologists often mark entire regions impacted by stroke, covering both affected and healthy tissues. Consequently, a model adept at normalizing pathological areas without altering healthy tissue tends to show a reduced overlap in these regions, leading to lower DICE scores. This issue is more apparent in cases with larger lesions, as indicated by the performance of AutoDDPM. These findings indicate a potential need to revisit current annotation and evaluation methods, particularly when the focus is on quantifying pathological burden.

Models with only moderate scores (Avg. 4) in normative learning metrics often demonstrate inconsistent diagnostic capabilities, hinting at overfitting to specific scenarios rather than wide-ranging disease detection. These models, with suboptimal scores in RQI, AHI, and

CACI, generally excel in limited contexts but lack broad applicability across diverse pathologies.

Less effective methods (Bottom 5) showed a significant negative correlation between restoration quality and anomaly detection performance. This finding aligns with existing literature[18,48], which notes that methods with dense latent spaces, often producing less sharp reconstructions, can surprisingly outperform more advanced methods. However, this trend may inadvertently lead researchers to focus on optimizing anomaly detection for a limited range of pathologies, typically marked by distinct features such as hyperintense or hypointense lesions. Pursuing this narrowed research direction risks developing models with limited adaptability and reduced clinical utility.

## Clinical validation
To evaluate the performance of different AI algorithms and validate our quantitative metrics−RQI and AHI−we conducted a comprehensive test involving 16 radiologists. Each was given 180 images in a randomized sequence, including 30 pathology-free originals and 30 from each AI method (15 from ATLAS and 15 from FastMRI+). Radiologists rated each image on 'Realness' (1 for likely fake to 5 for real), 'Image Quality' (1 for poor to 5 for excellent), and 'Health Status' (1 for pathological to 5 for healthy). Evaluating the CACI, which requires the analysis of input-reconstruction pairs, is challenging in a blinded test setting where radiologists view a randomized order of unknown images.

The violin plots in Fig. 5a reveal the scores given by radiologists for both real and AI-generated images. Even real images show score variability, especially regarding health status, indicating that not all pathology-free images should be automatically deemed 'healthy'. Differentiating between AI-generated and real images proves difficult, with real images scoring only marginally higher, highlighting AI's growing proficiency in replicating authentic radiological scans. The AutoDDPM model often deceived radiologists, receiving high 'Realness' scores (≥3). In contrast, the RA method achieved strong 'Health' scores, similar to real images, but displayed limited sample diversity, as reflected by its lower AHI score (0.16) compared to 0.49 for

## Table 1 | Anomaly Detection

| Era | Rank | Method | Avg. #det | Avg. F1 ↑ | Edema #det | Edema F1 ↑ | Mass #det | Mass F1 ↑ | Lesions #det | Lesions F1 ↑ | Absent Septum #det | Absent Septum F1 ↑ | Stroke Avg. | Stroke S | Stroke M | Stroke L | Rank |
|---|---|---|---|---|---|---|---|---|---|---|---|---|---|---|---|---|---|
|  | 14 | AE[18] | 25/171 | 2.17 | 0/18 | 0.00 | 0/26 | 0.00 | 1/22 | 2.27 | 0/1 | 0.00 | 3.43 | 0.39 | 3.82 | 13.20 | 14 |
|  | 11 | VAE[9] | 90/171 | 11.62 | 2/18 | 4.80 | 16/26 | 13.37 | 9/22 | 4.90 | 0/1 | 0.00 | 5.79 | 0.16 | 7.82 | 20.50 | 13 |
|  | 8 | LTM[8] | 110/171 | 16.83 | 4/18 | 11.48 | 19/26 | 16.97 | 10/22 | 6.07 | 0/1 | 0.00 | 7.25 | 0.71 | 6.78 | 19.73 | 11 |
|  | 9 | f-AnoGAN[34] | 99/171 | 11.70 | 2/18 | 3.44 | 16/26 | 12.36 | 8/22 | 3.68 | 0/1 | 0.00 | 8.35 | 1.27 | 10.38 | 28.16 | 9 |
|  | 13 | SI-VAE[36] | 82/171 | 9.67 | 0/18 | 0.00 | 12/26 | 7.08 | 6/22 | 3.97 | 0/1 | 0.00 | 10.84 | 1.71 | 14.10 | 34.75 | 8 |
|  | 4 | RA[16] | 142/171 | 39.73 | 12/18 | 45.56 | 21/26 | 30.78 | 17/22 | 29.50 | 1/1 | 15.38 | 14.54 | 2.79 | 19.93 | 42.29 | **5** |
|  | 7 | DDPM-G[19] | 114/171 | 14.56 | 5/18 | 9.07 | 17/26 | 12.01 | 11/22 | 5.32 | 0/1 | 0.00 | 8.24 | 2.82 | 9.58 | 23.20 | 10 |
|  | 5 | **DDPM-S[19]** | 133/171 | 23.09 | 14/18 | 35.51 | 22/26 | 12.42 | 16/22 | 16.92 | **1/1** | 14.29 | 17.77 | 3.50 | 23.97 | 52.32 | **3** |
|  | 10 | ceVAE[41] | 96/171 | 11.33 | 3/18 | 4.52 | 19/26 | 14.12 | 10/22 | 5.05 | 0/1 | 0.00 | 7.21 | 1.53 | 8.56 | 23.88 | 12 |
|  | 6 | MorphAEus[27] | 126/171 | 16.65 | 9/18 | 17.38 | 22/26 | 17.24 | 15/22 | 10.94 | 0/1 | 0.00 | 13.20 | 2.10 | 18.61 | 38.72 | 6 |
|  | 12 | MAE[42] | 84/171 | 12.38 | 2/18 | 4.63 | 14/26 | 13.66 | 7/22 | 3.71 | 0/1 | 0.00 | 12.16 | 2.98 | 15.21 | 36.83 | 7 |
|  | 3 | pDDPM[44] | 153/171 | 27.58 | **17/18** | **49.46** | 25/26 | 29.11 | 19/22 | 21.22 | **1/1** | 16.67 | 17.00 | 3.74 | 23.95 | 43.71 | **4** |
|  | 2 | **PHANES[45]** | 157/171 | 39.11 | 16/18 | 43.28 | 24/26 | 33.89 | **21/22** | 27.31 | **1/1** | 18.18 | **21.80** | 5.52 | **30.39** | **54.39** | **1** |
|  | 1 | **autoDDPM[40]** | **159/171** | **41.53** | 16/18 | 45.78 | **26/26** | **49.57** | **21/22** | **36.30** | **1/1** | **25.50** | 18.28 | **8.57** | 23.38 | 37.75 | **2** |

Comparative Performance of Generative Models in Anomaly Detection. The models are assessed based on the average number of detections and F1 scores for various pathologies on the FastMRI+ dataset, and the maximum [Dice] for different sizes of stroke lesions on the ATLAS dataset. Ranks (wrt. to mean performance) are provided for both tasks. Best results are shown in **bold**.

AutoDDPM. Enhancing reconstruction quality is crucial for methods like MAE, RA, and AnoDDPM to improve realism and diagnostic utility. Others, including AutoDDPM and pDDPM, should focus on better generating 'healthy' images, especially transforming large pathological areas. The primary challenge remains to balance accurate reconstruction with effective pseudo-healthy synthesis, a complex yet critical task in anomaly detection[40].

The heatmap in Fig. 5b shows the variability in realness scores among different radiologists. While the general trends from the previous violin plot are consistent, the scores vary between individual raters. Generally, real images received the highest realness scores. Except for the MAE method, which consistently received the lowest ratings, other AI methods varied in their ratings, with some instances even surpassing real images.

The bar charts in Fig. 5c highlight differences in evaluation scores between residents (N = 13) and experienced, board-certified radiologists (N = 3). These variations suggest that experience level affects the interpretation of image quality and health implications. Specifically, for realness scores (illustrated in the upper boxplot), experienced radiologists distinguished more clearly between real and AI-generated images. Additionally, the experts generally assigned higher ratings for both image quality and health status.

The correlation matrix in Fig. 5d demonstrates the relationships between the scores given by radiologists and our proposed metrics. The RQI showed a very high degree of correlation with the perceived image realness and quality. The AHI also exhibited a positive correlation with the perceived health status, albeit with slightly reduced strength. Minor inconsistencies in the AHI might stem from its use of the FID, which evaluates not only the "health" status of images but also other aspects such as sampling diversity and domain alignment between the evaluated sets.

## Discussion

Our analysis marks a significant shift in generative AI for medical imaging, advocating for emphasis on normative representation learning. To facilitate this, we introduced new metrics–Restoration Quality Index (RQI), Anomaly to Healthy Index (AHI), and Conservation and Correction Index (CACI)–designed to evaluate how well AI models learn the underlying, normal anatomy. Our findings demonstrate that the proposed metrics indicate the ability of a model to generalize across diverse conditions without relying on predefined labels or expectations about disease characteristics. We conducted a comprehensive clinical validation with 16 radiologists and found that the proposed RQI, and to a lesser extent the AHI, correlate well with clinical assessments.

Clinically, the implications are considerable. Models that master normative learning can discern subtle pathological nuances, essential for early disease detection and accurate diagnostics. Beyond improving diagnosis, the insights from this analysis can extend to pre-operative planning, therapeutic monitoring, and training healthcare professionals. The pseudo-healthy reconstructions provided by such models can serve as a 'baseline' view against which deviations can be accurately assessed. This could be particularly beneficial in complex scenarios such as brain volume loss. Here, the distinction between healthy and pathological states is often not clear-cut but exists on a continuum. The ability of these models to accurately represent this spectrum is invaluable for clinicians, aiding in more nuanced diagnosis and treatment planning. Ultimately, these AI models can assist clinicians in developing a deeper understanding of underlying pathophysiologies. This could be instrumental in generating novel hypotheses and advancing medical research, ultimately contributing to improved patient care strategies.

Despite the promising advancements, integrating these AI systems seamlessly into clinical workflows presents ongoing challenges. As critical tools in patient triage, these systems must exhibit

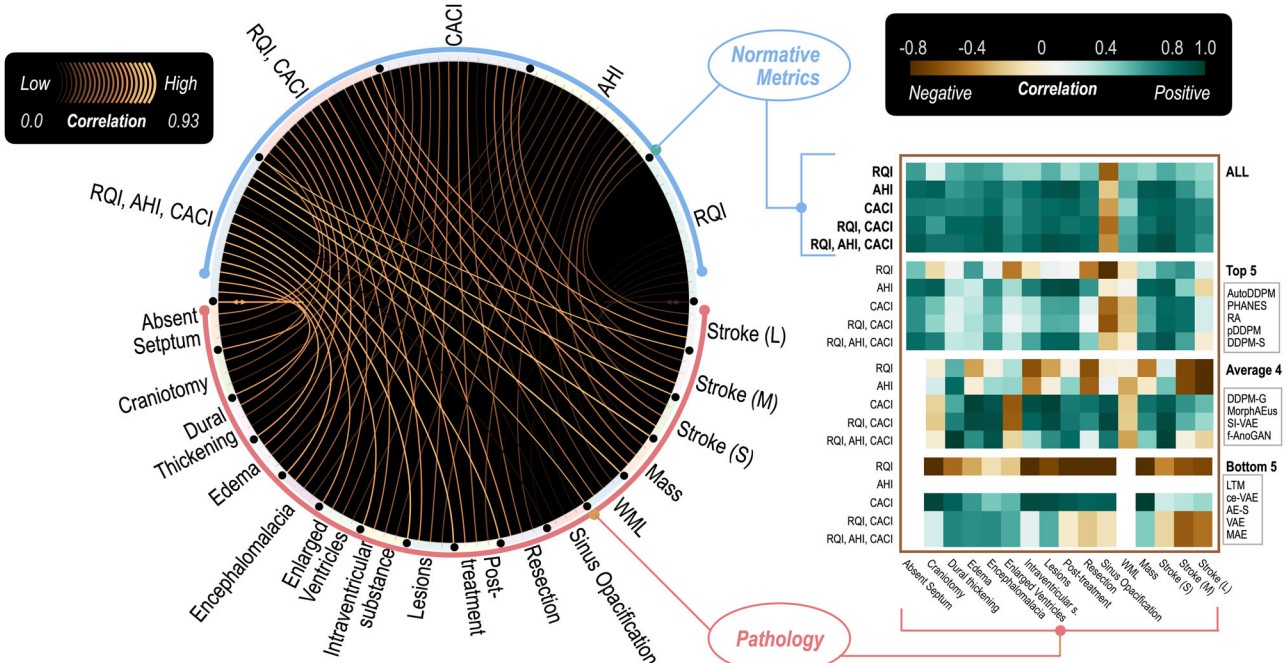

**Fig. 4 | Correlation between normative learning metrics and anomaly detection performance.** We present a chord diagram illustrating the correlations between normative learning metrics (RQI, AHI, CACI) and the detection performance of various pathologies. In the primary diagram, the width and brightness of the links denote the strength of correlation. Together, all three metrics show a strong positive correlation with improved anomaly detection performance across all methods. Heatmap insets on the right offer a detailed view of correlation patterns in different sub-groups (top-performing, average-performing, and low-performing methods). In some cases, intra-correlations within these groups may appear counter-intuitive, such as negative correlations with RQI for certain disease types. This suggests a complex optimization landscape where initial improvements in restoration quality may paradoxically yield inferior results, highlighting the nuanced challenge of refining anomaly detection techniques. Source Data are provided as a Source Data File.

robustness across various scanning protocols and adaptability to different patient demographics. Additionally, while 2D generative models produce high-quality slices, they often fail to maintain spatial integrity in a full 3D context[49]. Future work could evaluate these models across axial, coronal, and sagittal planes to assess their coherence in constructing accurate 3D structures.

In summary, our emphasis on normative learning and the introduction of new metrics contribute to developing AI models with greater clinical relevance. Moving forward, the development of AI models should emphasize not just excellence in experimental conditions but also resilience and adaptability within the complex realities of clinical environments. The pursuit of AI that comprehensively encompasses the range of normal anatomical variations is a critical step towards systems that enhance clinical decision-making and ultimately improve patient care and outcomes.

## Methods

### Background

This study complies with all relevant ethical regulations. The analysis was conducted on publicly available, anonymized datasets; no additional ethical approval was required. The comprehensive understanding of standard anatomical structures is imperative for the identification of anomalies. This process, as illustrated in Fig. 1, involves training models on datasets ($\mathcal{D}_H$) composed solely of images $\boldsymbol{x} \in \mathbb{R}^{C \times W \times H}$ from healthy subjects, where $C$, $W$, and $H$ represent the channels, width, and height of the images, respectively. These models are then employed to discern and morphologically convert pathological structures within mixed datasets ($\mathcal{D}_P$) into their normative (healthy) counterparts.

### Mathematical framework.

Generative models typically employ an encoder-decoder mechanism, where both the encoder $E$ and the decoder $G$ are formulated as neural networks with parameters $\boldsymbol{\theta}$. $E$ compresses an input image $\boldsymbol{x}$ into a lower-dimensional latent representation $\boldsymbol{z}$, typically in $\mathbb{R}^d$ where $d \ll C \times W \times H$. Then, a decoder (or generative model $G$) restores a pseudo-healthy image $\hat{\boldsymbol{x}} \subset \mathcal{D}_H$ from $\boldsymbol{z}$. The primary training objective is to minimize the reconstruction loss $L(\boldsymbol{x}, \hat{\boldsymbol{x}})$, often measured as the mean squared error (MSE), thereby optimizing the parameters $\boldsymbol{\theta}$. Interestingly, not all models follow this traditional "condensation" route. Diffusion models, for instance, take a different path. They preserve the original dimensionality and iteratively add noise to the input. This action allows the model to build up content while preserving intricate details[50].

### Anomaly scoring.

The discrepancy between the original image $\boldsymbol{x}$ from $\mathcal{D}_P$ and the reconstructed image $\hat{\boldsymbol{x}}$ from $\mathcal{D}_H$ is quantified as an anomaly score $S(\boldsymbol{x})$. This score provides a pixel-wise indication of anomalies, aiding in the localization of abnormal regions. For broader detection purposes, the anomaly presence in an image can be summarized by computing the mean or maximum anomaly score across all pixels: $S_{\text{image}} = \max_{i,j} S(\boldsymbol{x}_{i,j})$ or $S_{\text{image}} = \frac{1}{N} \sum_{i,j} S(\boldsymbol{x}_{i,j})$ where $x_{i,j}$ denotes the pixel at position $(i, j)$ and $N$ is the total number of pixels.

### Normative representation learning

**Req. i. Restoration Quality Index (RQI).** Evaluates restoration quality by semantically comparing two images using the Learned Perceptual Image Patch Similarity (LPIPS). LPIPS is a metric that quantifies perceptual similarity between images in a way that aligns closely with human judgment. This metric is derived from the internal activations of deep convolutional networks, which, although trained on high-level classification tasks, have shown to be effective in representing perceptual similarity[51]. The RQI evaluates the fidelity of image restorations of unseen healthy samples, with a lower LPIPS indicating greater

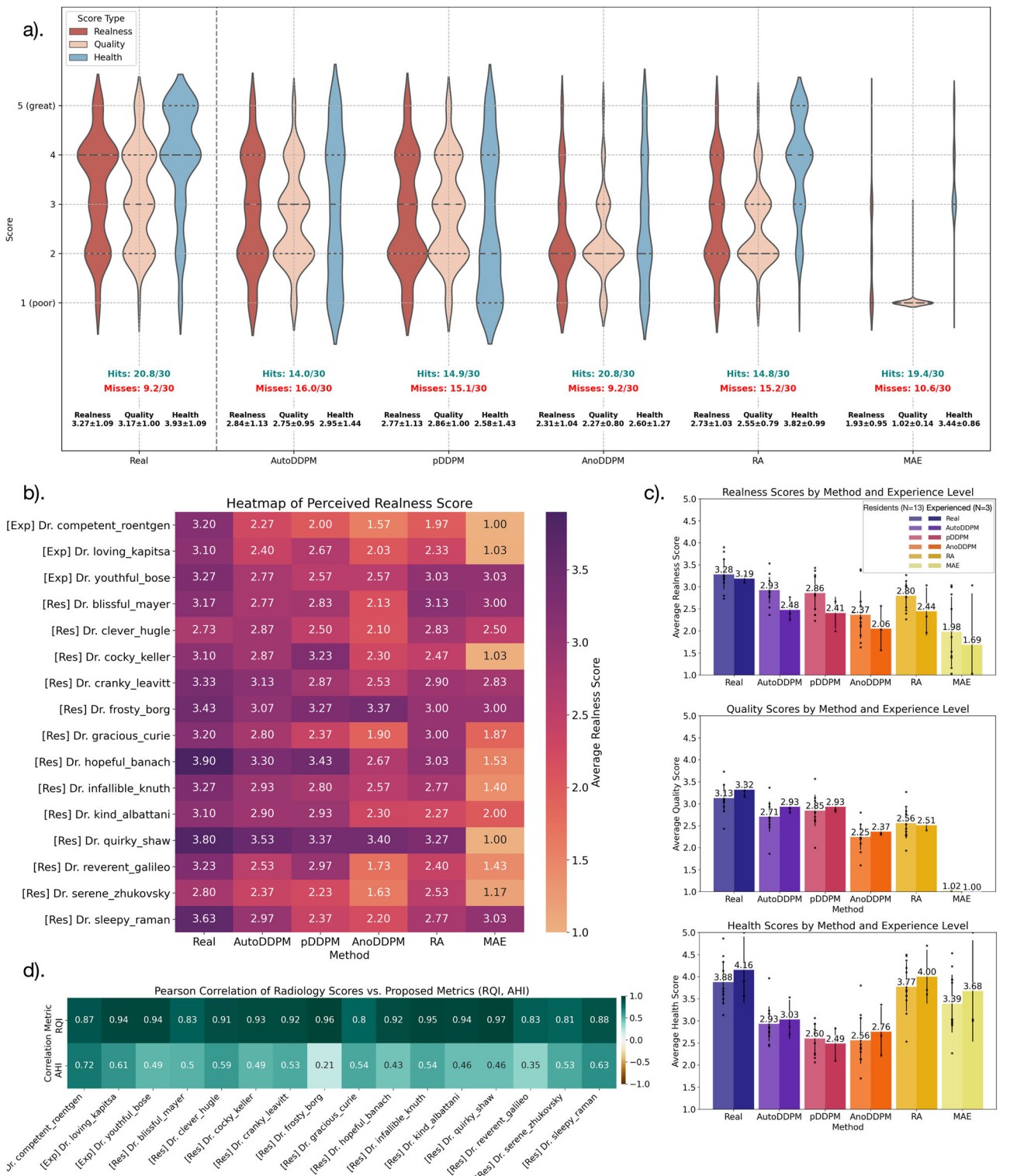

**Fig. 5 | Radiologists in a randomized study performed a test to differentiate real and AI-generated images. a** Violin plots display average score distributions across metrics like Readiness, Quality, and Health. **b** Heatmap indicates inter-rater variability in scoring image realness among radiologists, including both experienced ([Exp]) and resident ([Res]). **c** Bar charts compare mean scores between residents ($N = 13$) and experienced radiologists ($N = 3$). The error bars represent the standard deviation (SD) across raters. Individual data points are overlaid to show the score distribution. **d** The matrix shows Pearson correlation coefficients between proposed metrics (RQI, AHI) and radiologists' judgments of image quality and health. Source Data are provided as a Source Data File.

accuracy:

$$\text{RQI} = \frac{1}{N} \sum_{\boldsymbol{x} \in \mathcal{D}_\text{H}} 1 - \frac{\text{LPIPS}(\boldsymbol{x}, \hat{\boldsymbol{x}}) - \min_{lp}}{\max_{lp} - \min_{lp}}, \quad (1)$$

where $\mathcal{D}_\text{H}$ represents the dataset of $N$ unseen healthy samples. The $\min_{lp}$ denotes the minimum LPIPS possible, which is 0 in the case a method returns the identity function. We set $\max_{lp}$ to 25, which is higher than the worst-performing method (VAE). To maintain consistency and avoid negative values, performances worse than 25

can be capped at this maximum threshold, thus normalizing the metric between 0 and 1.

**Req. ii. Restoration to a healthy state.** To test how well the models can normalize image abnormalities, we propose measuring the distances between image distributions. Specifically, the Anomaly to Healthy Index (AHI) measures the ability to transform a pathological dataset $\mathcal{D}_P$ towards the normative patterns of the healthy training set $\mathcal{D}_H$. Using the Fréchet Inception Distance (FID)[52] to measure the distance between two distributions, we compute AHI as:

$$\text{AHI} = \max\left(0, 1 - \frac{\max(0, \text{FID}_{\text{RP,H}} - \text{FID}_{\text{UH,H}})}{\text{FID}_{\text{P,H}} - \text{FID}_{\text{UH,H}} + \epsilon}\right), \qquad (2)$$

where $\text{FID}_{A,B}$ represents the FID between two datasets $A$ and $B$, and P and RP represent the pathological and restored pathological samples, respectively. AHI is normalized to the range between 0 and 1.

**Req. iii. The healthy conservation and Anomaly Correction Index (CACI).** Utilizes the Structural Similarity Index (SSIM)[53] to assess the proficiency of the models in preserving healthy tissue details and correcting anomalies:

$$\text{CACI} = \frac{2}{\left(\frac{1}{\text{SSIM}_H + \epsilon}\right) + \left(\frac{1}{\max(0, \text{SSIM}_H - \text{SSIM}_{AN}) + \epsilon}\right)}, \qquad (3)$$

where $\text{SSIM}_H$ measures similarity within healthy regions and $\text{SSIM}_{AN}$ within anomalous regions. This index requires lesion segmentation masks annotated by expert clinicians, formally defined as binary masks $m_i$, where $m_i = 1$ if the pixel is identified as anomalous and $m_i = 0$ otherwise. CACI is bounded between 0 and 1.

**Metric Integration.** Integrating the RQI, AHI, and CACI offers a comprehensive evaluation of generative models in medical imaging. However, individual assessments of these metrics still provide unique insights into specific aspects of model performance, highlighting strengths and areas for improvement. For a balanced overall evaluation, we propose a combined metric calculated as:

$$\text{RQI, AHI, CACI} = \frac{2 \times \left(\frac{\text{RQI} \times \text{CACI}}{\text{RQI} + \text{CACI}}\right) + \text{AHI}}{2}. \qquad (4)$$

**Datasets**
We used the following datasets in our manuscript:

**Healthy data for training.** Two public T1w brain MRI datasets from healthy individuals, $\mathcal{D}_H$, were employed for model training: IXI (581 training samples) and FastMRI+[46] (131 training, 15 validation). We kept 30 samples from the FastMRI+ as unseen healthy test samples, $\mathcal{D}_{\text{UH}}$.

**Pathology data for evaluation.** We used two public datasets containing several disease classes as our datasets, $\mathcal{D}_P$, containing pathology:

FastMRI+ Dataset[46]: It includes 643 annotated pathologies across 30 classes. We selected mid-axial T1-weighted slices, yielding 171 unique pathologies in 13 classes. We used the annotations provided as bounding boxes by medical experts to assess the detection ('#det') and precision (F1 score) of the models[16]. We considered a detection to be a true positive (TP) if at least 10% of the pixels within the annotated bounding box were flagged as anomalous. False positives (FP) were calculated as the ratio of misdetected pixels on healthy tissue relative to correctly detected pixels within the anomaly box. Finally, we report the F1 score as: $F1 = \frac{1}{N}\sum_{i=0}^{N} \frac{2 \times P \times TP}{P + TP}$, where $P = \frac{TP}{TP+FP}$ and $N$ is the number of test cases.

ATLAS v2.0 Dataset[47]: Featuring scans with stroke lesions, the ATLAS v2.0 presents a challenging range of lesion sizes and intensities. The dataset includes 655 training scans with detailed annotations, offering in-depth views of stroke anomalies. We stratified the test sets into small (first 25th percentile, < 71 pixels), medium, and large lesions (top 25th percentile, ≥570 pixels) for performance evaluation, with the largest Dice coefficient (⌈Dice⌉) as the metric. ⌈Dice⌉ represents the theoretical maximum segmentation accuracy, achieved by a greedy search for the best residual threshold on the test set[8]. We excluded the middle slices containing no visible anomalies (N = 215) and the scans showing visible unlabeled artifacts (N = 20) as in ref. 40.

**Data pre-processing.** We have intentionally preserved the variability inherent in the data, adhering to the preprocessing protocols of the original datasets (See Supplementary Table 2 for details). Additionally, we normalized the mid-axial slices to the 98th percentile, applied padding, and resized them to a resolution of 128 × 128 pixels. For training, we used affine augmentations with a random rotation up to 10 degrees, up to 0.1 translation, scaling from 0.9 to 1.1, and horizontal flips with a probability of 0.5.

### Reporting summary
Further information on research design is available in the Nature Portfolio Reporting Summary linked to this article.

## Data availability
All datasets used in this study are publicly available, and their usage complies with the respective terms and conditions of the databases where they were sourced.

The FastMRI dataset is available at https://fastmri.org, with the respective labels of the fastMRI+ dataset at https://github.com/microsoft/fastmri-plus. The FastMRI dataset is provided under a Dataset Sharing Agreement by NYU Langone Health, permitting its use for internal research or educational purposes only. Our use of this dataset strictly adheres to these terms, as it has been employed solely for non-commercial academic research purposes.

The IXI dataset is available at https://brain-development.org/ixi-dataset/and is licensed under the Creative Commons CC BY-SA 3.0 license. This license permits use, distribution, and adaptation of the data with appropriate credit and indication of changes. We have ensured compliance with these requirements by properly citing and crediting the dataset in this study.

The Atlas 2.0 dataset is available at https://atlas.grand-challenge.org. The terms of this dataset require acknowledgment of its source in publications or presentations. We confirm that the Atlas 2.0 dataset has been used in accordance with these terms and appropriately cited in this manuscript. Source data are provided with this paper.

## Code availability
The implementation of all models and code presented in this manuscript are publicly available at https://github.com/compai-lab/2024-ncomms-bercea.git[54].

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

## Acknowledgements

The authors thank Prof. Jan Kirschke, Philipp Raffler, Dr. Cornelius Berberich, Dr. Kirsten Jung, Dr. Su Hwan Kim, Dr. Lukas Walder, Dr. Severin Schramm, Dr. Joachim Schmitt, Constanze Ramschuetz, Lena Schmitzer, Dr. Olivia Kertels, and Mirjam Beyrle for their invaluable contributions to our multi-reader study. Their expertise and dedication were essential in validating our proposed metrics and enhancing the rigor of our work. C.I.B. is funded via the EVUK program ("Next-generation AI for Integrated Diagnostics") of the Free State of Bavaria and partially supported by the Helmholtz Association under the joint research school 'Munich School for Data Science'. This work was in part supported by Berdelle-Stiftung (grant TimeFlow).

## Author contributions

C.I.B. led the project; contributed to the conception of key ideas and study design; conducted experiments; interpreted data and results; and drafted and edited the manuscript. J.A.S. and D.R. secured funding for the project; provided critical feedback on study design, data analysis, and interpretation; and contributed to manuscript editing. B.W. led a clinical study involving 16 radiologists to validate the proposed metrics; provided clinical insights; and reviewed the manuscript for clinical accuracy. All authors reviewed and approved the final manuscript.

## Funding

## Competing interests

The authors declare no competing interests.
