## [Transparent Peer Review file · Nature Communications]

Evaluating Normative Representation Learning in Generative AI for Robust Anomaly Detection in Brain Imaging

Corresponding Author: Mr Cosmin Bercea

Version 0:

Reviewer comments:

Reviewer #1

(Remarks to the Author)

The authors present a score that evaluates the quality of simulated brain scans obtained by generative algorithms. They observe that their score(s) are able to rank simulations and simulation results in a very meaningful way, i.e., ranking those at the top that are visually most appealing and - more importantly - that are identifying outlying pixels in a very consistent fashion. Overall, I feel this is a very important paper, I like the idea of the scores offered here, and the experiments that systematically compare properties of different generative algorithms (which is only enabled through these new scores) are much appreciated. The scores do fill a void in an emerging domain in medical imaging AI!

Below I summarize some observations that may help the authors to improve the presentation of their idea.

== General comments ==

* The contribution, i.e., the scores that are introduced, should be motivated, introduced, and explained better. (I do understand that the Nature format of presenting Intro and Results first, and any Methods later is awkward for introducing ideas and concepts, but the authors should double check that readers can follow the explanation without jumping back and forth between intro and methods.) The same is true for terms like "normative learning" that made it into the title - some description and definition would be helpful. How does it relate to the metrics proposed here? Personally, I was wondering whether "normative learning metrics" refers to metrics that are used for "normative learning" (whatever that may be or whatever this term may mean to the authors), or to "learning metrics" that happen to have some normalization included. I would be guessing the latter is true, but I am not sure. It would be nice if no guesswork is left to the readers.

* In the same way, a serious attempt to formalize the presented scores should be made. Right now it is not clear what is assumed to be available (images? image segmentations? simulated/generated scans? partially, i.e., conditionally simulated/generated scans?), and critical details of the score remain hidden in references that are left to the reader to look up.

* I like the observation that the higher the score, the better the reconstruction of the algorithm. But if all three correlate well (i.e., perfectly), it would be ok to just calculate one and forget about the other two? The authors should make a proposal about how to fuse them. I read the word "integrate" somewhere in the text, but maybe an equation would do it that gives rise to a "integrative normative learning scores".

* The authors somehow hypothesize that generative models would offer better capabilities for modelling anatomical images than anything before - and I would somehow agree - which is why they are presenting no other baselines for the evaluation tasks at hand. Still, a critical voice could argue that a pixel-wise estimation of outlyingness of the intensity of a tissue will get you as far in identifying abnormal pixels in Fig. 2, as the scores evaluated here. These probabilistic generative models (NB: this is a different "generative") would be at least in the example of brain anomaly detection a relevant classical baseline. Maybe the authors can inform us how well they do in terms of your scores?

* I am a bit concerned about the generalization of the presented results beyond the brain. The brain itself is rather easy to interpret, there are well-defined homogenous regions, and rather regular anatomy where anatomical differences and diseases are recognised from local intensity changes. (See my comment before.) The authors simplify this problem by

somehow pre-aligning the scans (I assume?) and only inspecting and evaluating a single well-defined slice of the whole 3D volume. I wonder: are the scores applicable to synthetically generated 3D data at all? And to more complex diagnostic tasks? (I believe they are?) Are the scores identify shortcomings of 2D generative algorithms that typically look nice in every single axial slice, but not so much when inspected under a 90 degree angle. I.e., how would a 3D version of these scores rank a stack of individually synthesised 2D images vs. a stack obtained from an algorithm generating 3D images? I understand that means for generating 3D data are still limited, but it would be nice to have at least a discussion of the 2D related shortcomings added to the paper. Similarly, it would be good to have discussion of application domain related shortcomings, and to learn from the authors about the potential for a generalization to other situations, like the anomaly detection in abdominal CT. Right now this is a paper about anomaly detection in neuroimage data. (BTW: the title does refer to the limitations that is "image data" at all, which even exaggerates to the contrast to the brain image anomaly experiments.)

Please find below some more specific pointers about the aforementioned matters.

== To the authors: ===

* I would agree with all that is said in the intro. But is it worth to introduce (to define?) what normative learning is in the intro as well?

* p2 There is quite some prominence for the Generative AI algorithms (which is all good). But, they are no more than means for introducing and justifying the novel scores introduced here. Can you introduce the score you propose at the same level as the generative AI algorithms? Offering some high level insights or intuition? At present, the metrics are introduced in a part of a paragraph in the results section.

* The experiments are exclusively about the brain, a rather well structure organ, and mostly about local intensity. Does any of this apply to topological chances as well? (Think about vascular structures: bifurcation vs. trifurcation.) What would these scores look like for an abdominal CT?

* Fig 2, and elsewhere: Given that your anomaly detection heavily relies on intensity differences - can you add a intensity outlyingness estimation to your comparisons? In your example of brain anatomy you would get rather far with classical generative probabilistic models that model anatomy with atlas templates and estimate local tissue-specific outlyingness by calculating tissue-specific intensity distributions. Can you compare? (I don't ask you to compare against this specific method, but just to give you a pointer to the body of literature I refer to: <https://pubmed.ncbi.nlm.nih.gov/11513020/>.)

* p 4, l 143: I understand the details are in the method section, but can you mention the numbers of the cases (approximative) and the number of different test cases?

* Fig 2: I am not sure this figure is a good summary. What is the brain visualized for?

* p 5, 172: You describe your three metrics and mention that they need to be holistically - and qualitatively - interpreted to provide a comprehensive overview. As this is about metrics and expressing diffuse statements about quality into simple and quantitative representations I wonder: can you offer some summary statistic that merge the three scores?

* p 6, l 238: what is "balanced" normative learning? You are also mentioning the integration of the three scores three lines earlier. Is this a plain numeric summation you are recommending?

* p 7, 305: "designed to evaluate the deep under306 standing of normal anatomy" => ?? rephrase ?

* p 8, 383: LPIPS is not defined. As a reader I don't like to look up this reference 47. I understand that your contribution is in the proposal to normalize this score, but can you offer those of your readers who are unwilling to access the paper by Zhang et al a mathematical definition (or, in case this is difficult, a brief summary of how this score is calculated)?

* p8, l 397: So this score requires having a segmentation of the lesion/anomaly? But if I have a fully annotated test set, then I could go for an evaluation of false positives or negatives and even segmentations scores, couldn't I? Maybe it would help if you would explicitly describe (formalize?) how you foresee that your scores are employed.

* p 8, l 404. So you have about 700 datasets (minus the 235 that you comment on on the next page). You only use one 2D slice for the extraction. Do you do any preprocessing like alignment to restrict the variability of the appearance of these 700 slides? A rigid or affine registration of the 3D volumes? Or a nonlinear one ? (I hope not!) Also, please report physical dimensions - pixels are about 1 mm² in size?

(Remarks on code availability)

Reviewer #2

(Remarks to the Author)

Brief Summary:

Normative learning is to train AI models on large datasets of typical images from healthy volunteers. Once normal models are created, then anomaly detection can be conducted. Traditional methods do not take into account (much) normative

learning. In this article, the authors introduce new metrics to evaluate normative AI models. Authors apply these metrics to several generative AI frameworks including the recently hot topics of advanced diffusion methods, and then test the application of brain pathologies. Some interesting and good results were obtained.

Strengths:

This study addresses an interesting and a significant topic of anomaly detection with generative AI methods from the evaluation perspective with new metric.

Normal T1W brain images were used for experiments (IXI and fastMRI datasets), and for evaluation enhanced FastMRI and ATLAS v2.0. This strategy allows authors to test the capabilities of different models in detecting and localizing different anomalies.

The authors present three key metrics: RQI, AHI, CACI. These metrics offer some nice perspectives for the generative models.

Weaknesses:

Introduction is written nice but to be honest I am sensing little bit chatGPT kind of senses, which I am not against, but look like similar to currently written many papers.

Figure 1 has some styles which are not understandable. It looks like a magazine cover, rather than a scientific paper figure.

It seems these three metrics (proposed) should be evaluated together, rather than in isolation (as authors themselves approve), but there is no clue if one single metric can be derived from these three still satisfying important properties like between 0-1, and somehow probabilistic and etc.

There are many unjustified sentences and words like profound etc...I do not think the study implies such extreme excellence yet, maybe initial results with good metrics, but they do not show clinical implications yet.

Not sure how these metrics are affected from preprocessing steps? MRI has a lot of challenges to address before the actual processing.

Potential biases coming from different networks when FID is used are not discussed.

AHI includes FID, and FID is not capturing semantic similarity, but feature similarity. Therefore, I am not sure if AHI is really capturing the semantic similarity. More elaboration are needed.

There seems to be no expert evaluation other than some elaboration on certain scores. Why not run a Turing test with radiologists (multiple?)

(Remarks on code availability)

code will be provided (they say).

Reviewer #3

(Remarks to the Author)

In this article, the authors present new metrics to evaluate deep generative models for unsupervised anomaly detection (UAD) in medical images. These metrics assess how well deep generative models learn the healthy distribution, as opposed to their ability to detect anomalies, as is done usually. The metrics are used to evaluate about fifteen approaches in the context of brain lesion detection in structural MRI.

Designing new ways to evaluate UAD is crucial to understand how these approaches actually work and fully assess their potential. The paper is well written and convincing, but I would have appreciated a deeper analysis of the metrics. In particular, it would be interesting to better know how they behave depending on the severity of the lesions. These could be done pushing further the evaluation with the stroke dataset or using simulated data. I also have several specific comments, see below.

Detailed comments

- I.35: 'The true innovation of Generative AI lies in normative learning, yet its evaluation often skews towards anomaly detection, leading to biased or incomplete assessments.'

I would reformulate this sentence as it gives the impression that evaluating an approach (normative learning) using the task it was developed for (anomaly detection) is a bad thing. It is not, but it should not be the only evaluation criterion.

- I.39: 'certain approaches optimize solely for hyper-intense lesion detection [18]'

Could you be more specific, i.e. which approach(es) from all the ones being compared in [18] are you targeting?

- I.41: 'anchor their design principles around large hyper intense tumors [19]'
In this approach, training is also performed using healthy subjects only. Please explain why you think it is specific to a type of lesion.

- I.42: 'resort to approximating the unknown anomaly distribution through self- or weakly-supervised strategies [20, 21].'
Do you imply that this is a bad thing? If so, why?

- Fig. 2. Not having the AE displayed is a bit confusing, especially as in Table 1 the letter A refers to AE.

- I.85: 'reconstruction quality, which is essential for precise anomaly detection.'
Some works seem to imply something different, e.g. Lambert et al., ISBI 2021. As it is a point that you raise later in the result, you could moderate it here and insist on it later.

- Fig. 3. It might be useful to specify that the normative learning metrics were obtained for the whole dataset and do not correspond to the exemplar image being displayed. It would be especially useful as a lesion is visible in the AutoDDPM reconstruction while the AHI is very high, while for example no lesion seems visible in the DDPM-G reconstruction but the AHI is low.

- I.154: Following-up on the previous comment, it would be good to clarify whether normative learning metrics are computed for a whole dataset or can be obtained per image.

- Table 1: How were the ranks obtained, i.e., based on what criteria? This information is important as later in the text (l. 200–209) the ranks are commented upon, but if all the methods have very similar scores, then the rank may not mean much.

- Fig. 4: It is not easy to see which line connects which metric/task. Also, one line, starting on the left and going in a horizontal direction, seems to stop before reaching another point.

- I.277: 'hinting at overfitting to specific scenarios rather than wide-ranging disease detection.'
How do you explain this behaviour, knowing that these approaches, as the others, are trained on healthy data?

- I.353: Almost nothing is said about the generative models being evaluated (2D/3D, architecture, hyper-parameters, losses, training procedure, etc.). Even though these are existing models, a minimum should be said, e.g. whether everything was kept the same as in the paper being cited; if modifications were made, which ones.

- I.403: Similarly, little is said about the data. For example, no demographic information is provided, the MRI sequence is barely mentioned, it is not clear whether training is done with all 2D slices or just mid-axial slices.

- I.431: How are the residual images processed to compute the DSC?

(Remarks on code availability)

I haven't tried running the code but many key aspects seem present (detailed README, requirements, proper organisation, etc).

Version 1:

Reviewer comments:

Reviewer #1

(Remarks to the Author)

Thank you for the responses. They helped me in clarifying about the paper. This is a very nice study that can be impactful in our community!

(Remarks on code availability)

Reviewer #2

(Remarks to the Author)

Adding several radiologists to do user study is great, the paper becomes very solid, and field will get benefit. Thanks for adding this component.

Some further minor adjustments are necessary:

-- supplementary table 1, all the references are missing

-- figures and tables are nice, are they reproducible from code ? or powerpoint/paint ?

(Remarks on code availability)

- compared models are included
- not sure if everything is there but I have seen several necessary code, did don't run it myself.
- there is a readme file explaining how to close and run the main file.

Reviewer #3

(Remarks to the Author)

The authors have carefully revised their manuscript, clarifying several points and adding a clinical validation involving radiologists, overall strengthening the work. However, I still have comments, see below.

Detailed comments

- I.155 'This approach marks a shift in the assessment of generative AI, positioning it not just as a tool for disease identification but as a system for understanding the spectrum of human health in medical imaging.' The last part of the sentence appears like an overstatement not supported by results, please rephrase.
- Several acronyms are not defined, e.g. MAEs (l.120), LTMs (l.154).
- I.173 'To fuse the metrics, we propose a harmonic mean between RQI and CACI, averaged with AHI.' This is a good thing to propose a way to combine the three metrics proposed but is this composite metric evaluated in the present work or some results presented? It does not seem to be the case. Please clarify and/or provide some results obtained with this composite metric to justify its soundness.
- I.234 'However, the most comprehensive insights are obtained when these metrics are collectively analyzed, with combinations such as CACI and RQI, and particularly the integration of all three metrics (RQI, AHI, CACI), demonstrating enhanced predictive anomaly detection power.' Is the composite metric used in this analysis?
- Fig. 4: Please clarify the metric used to assess the 'detection performance'.
- I.299 'To evaluate the performance of different AI algorithms and validate our quantitative metrics—RQI and AHI—we conducted a comprehensive test involving 16 radiologists.' The authors could explain here why the CACI is not part of the evaluation (instead of having to wait until the end of the section).
- Fig. 5 (a): What does the width of the violin plot correspond to? Please comment on the appearance of the plots for the MAE.
- I.420 'Additionally, while 2D generative models produce high-quality slices, they often fail to maintain spatial integrity in a full 3D context [49]. Evaluating these models across axial, coronal, and sagittal planes helps assess their coherence in constructing accurate 3D structures, enhancing the utility of AI in complex diagnostic tasks.' This is a very generic statement that seems disconnected from the present work. Please rephrase to clearly present it as a limitation or future work as this analysis is not performed here.
- I.497 Several data sets are mentioned when defining the metrics (D_{UH} , D_P , D_H). It would be good to specify what data from the IXI, FastMRI+ and ATLAS datasets are used to create them as it is currently unclear.
- I.541 I insist that more should be written about the datasets (e.g., a summary table in the supplementary). It is commendable to release code, but it is not the job of the reader to check code, articles and websites to see whether there are large discrepancies in population demographics or image characteristics (field strength, sequence (MPRAGE? SPGR?), etc.).
- I.552 'We used the annotations provided as bounding boxes by medical experts to assess the detection ('#det') and precision (F1 score) of the models [16].' This procedure should be described in greater details here (in the supplementary).
- I.565 When mentioning 'the largest Dice coefficient' please provide more details and/or add a reference.
- I.569 The pre-processing performed, even if run by the dataset providers, should be explained in the supplementary. Again, this is not the readers' job to fish for this information within multiple sources.

(Remarks on code availability)

Evaluating Normative Representation Learning in Generative AI for Robust Anomaly Detection in Brain Imaging

– Point-by-point response –

Cosmin I. Bercea^{1,2*}, Benedikt Wiestler^{1,3}, Daniel Rueckert^{1,3,4}, Julia A. Schnabel^{1,2,5}

¹Technical University of Munich, Munich, Germany.

²Helmholtz AI and Helmholtz Center Munich, Munich, Germany.

³Dept. of Neuroradiology, Klinikum Rechts der Isar, Munich, Germany.

⁴Imperial College London, London, UK.

⁵King's College London, London, UK.

*Corresponding author(s). E-mail(s): cosmin.bercea@tum.de;

We thank the reviewers and editor for their time and constructive feedback, which has significantly improved the quality of our work. In response to the feedback, we have made the requested updates to the manuscript:

- We conducted a comprehensive, multi-reader randomized study with 16 neuroradiologists to compare our proposed metrics against expert evaluations. This on its own already forms a substantial new contribution to the field, and we are particularly grateful to Reviewer 2 for making this thoughtful suggestion. The key result of this study reveals that our proposed metrics—the Reconstruction Quality Index (RQI) and Anomaly to Healthy Index (AHI)—show strong correlations with the radiologists’ evaluations of realness, image quality, and health status. Additionally, our findings indicate that AI models can effectively simulate real images, frequently leading radiologists to mistakenly assess them as genuine.
- In response to Reviewers 1 and 3’s inquiry about the challenges of detecting brain anomalies, we have enhanced the supplementary materials with comprehensive details on the anomaly detection task. This includes a detailed table with results on all disease classes (Supplementary Table 1), an intensity distribution plot to illustrate the complexity of differentiating between healthy and pathological samples (Supplementary Fig. 1), and qualitative examples that showcase how AI methods can accurately address complex structural pathologies, such as enlarged ventricles, the mass effects of tumors, and atrophic changes following strokes (Supplementary Fig. 2).
- We adapted the title and scope of our manuscript as suggested by Reviewer 1: “Evaluating Normative Representation Learning in Generative AI for Robust Anomaly Detection in Brain Imaging”.
- We revised the manuscript to incorporate any more minor reviewer comments, thereby further enhancing the clarity and quality of our work.

We have carefully addressed each review comment below (we highlight **removed** and **added** text in color):

Reviewer 1:

R1.1: “The authors present a score that evaluates the quality of simulated brain scans obtained by generative algorithms. They observe that their score(s) are able to rank simulations and simulation results in very meaningful way, i.e., ranking those at the top that are visually most appealing and - more importantly - that are identifying outlying pixel in a very consistent fashion. Overall, I feel this is a very important paper, I like the idea of the scores offered here, and the experiments that systematically compare properties of different generative algorithms (which is only enabled through these new scores) are much appreciated. The scores do fill a void in an emerging domain in medical imaging AI!”

A.1.1: We appreciate your positive feedback and recognition of the effectiveness of our proposed scores in evaluating simulated brain scans and consistently identifying outlying pixels. We are grateful for your

insights, which have notably enriched this manuscript.

**R1.2:** “The contribution, i.e., the scores that are introduced, should be motivated, introduced, and explained
better [...] The same is true for terms like ”normative learning” [...]”

**A.1.2:** We thank the reviewer for this great suggestion. We have now refined the terminology in our
manuscript by changing the name to ”normative representation learning” in the title and throughout the
text. We have also enhanced the introduction with a concise explanation of this concept: “[...] *in normative
representation learning, a concept driven by data to uncover concise characteristics of a healthy population.*”
Additionally, a new subsection introduces and explains our metrics, focusing on their role in assessing the
ability of AI methods to learn anatomical patterns from control patients and generate pseudo-healthy images
(lines 69-87 in the revised manuscript):

“We propose specific metrics to assess the quality of the pseudo-healthy restorations:

*Restoration Quality Index (RQI): is an image-based metric that evaluates the perceived quality of
synthesized images by measuring their semantic similarity to original inputs.*

*Anomaly to Healthy Index (AHI): measures the closeness of the distribution of restored pathological
images to a healthy reference set.*

*Healthy Conservation and Anomaly Correction Index (CACI): measures the effectiveness of models in
maintaining integrity in healthy regions and correcting anomalies in pathological areas.*

*These metrics collectively provide a more comprehensive evaluation, extending beyond simple anomaly
mapping to include assessments of the quality and accuracy of the normative and pseudo-healthy represen-
tations generated by AI models.”*

**R1.3:** “In the same way, a serious attempt to formalize the presented scores should be made. Right now
it is not clear what is assumed to be available (images? image segmentations? simulated/generated scans?
partially, i.e., conditionally simulated/generated scans?), and critical details of the score remain hidden in
references that are left to the reader to look up.”

**A.1.3:** We now have further clarified the data inputs and scoring methodology in the revised Methods
section (lines 506-624 in the revised manuscript), within the page limitations. We have included explicit
details on the types of data used and a more comprehensive description of the scoring process. For instance,
the input images and the generated scans are described as: “*This process, as illustrated in Figure 1, involves
training models on datasets (\mathcal{D}_H) composed solely of images $x \in \mathbb{R}^{C \times W \times H}$ from healthy subjects, where C ,
W , and H represent the channels, width, and height of the images, respectively. These models are ~~then tasked~~
~~with recognizing and replacing~~ then employed to discern and morphologically convert pathological structures
~~in within~~ mixed datasets (\mathcal{D}_P) ~~with healthy-like~~ into their normative (healthy) counterparts. [...] Generative
models typically employ an encoder-decoder mechanism, where both the encoder E and the decoder G are
formulated as neural networks with parameters θ . An encoder E compresses an input image x into a lower-
dimensional latent representation z , typically in \mathbb{R}^d where $d \ll C \times W \times H$. Then, a decoder (or generative
model G) restores a pseudo-healthy image $\hat{x} \in \mathcal{D}_H$ from z .”*

For CACI, additional segmentation labels are needed to compute the score, there we define these as: “*This
index requires lesion segmentation masks annotated by expert clinicians, formally defined as binary masks
m_i where $m_i = 1$ if the pixel is identified as anomalous and 0 otherwise. CACI is bounded between 0 and 1.*”

**R1.4:** “I like the observation that the higher the score, the better the reconstruction of the algorithm. But
if all three correlate well (i.e., perfectly), it would be ok to just calculate one and forget about the other two?
The authors should make a proposal about how to fuse them. I read the word ”integrate” somewhere in the
text, but maybe an equation would do it that gives rise to a ”integrative normative learning scores.”

**A.1.4:** Thank you for your insightful comment. Although our metrics correlate strongly, each assesses
unique aspects of learning normal anatomy. We have originally shown in Figure 4, that combining the
metrics leads to a much stronger correlation to anomaly detection performance. As suggested, we have
further clarified and formalized the integrative approach to combine these metrics in the Sections ”Normative
Learning Evaluation” (lines 233-238) and ”Methods” (lines 617-624) of the revised manuscript: “*To fuse the
metrics, we propose a harmonic mean between RQI and CACI, averaged with AHI. This approach balances
image quality and anomaly correction while mitigating the impact of near-zero AHI scores on the overall
evaluation (see Equation 4.)*”

“*Integrating the RQI, AHI, And CACI offers a comprehensive evaluation of generative models in
medical imaging. However, individual assessments of these metrics still provide unique insights into specific
aspects of model performance, highlighting strengths and areas for improvement. For a balanced overall
evaluation, we propose a combined metric calculated as:*”

$$RQI, AHI, CACI = \frac{2 \times \left(\frac{RQI \times CACI}{RQI + CACI} \right) + AHI}{2}. \quad (1)$$

R1.5: “The authors somehow hypothesise that generative models would offer better capabilities for modelling anatomical images than anything before - and I would somehow agree - which is why they are presenting no other baselines for the evaluation tasks at hand. Still, a critical voice could argue that a pixel-wise estimation of outlyingness of the intensity of a tissue will get you as far in identifying abnormal pixels in Fig. 2, as the scores evaluated here. These probabilistic generative models (NB: this is a different “generative”) would be at least in the example of brain anomaly detection a relevant classical baseline. Maybe the authors can inform us how well they do in terms of your scores?”

A.1.5: We appreciate your insights regarding anomaly detection methods. Our study primarily focuses on evaluating synthesized images through generative AI models and not directly on pixel-wise anomaly detection techniques, which fall outside our current scope. However, to address your concerns and illustrate the challenges involved, we have included a new analysis in Supplementary Fig. 1. This figure compares the distribution of intensities between healthy tissue and anomalous pixels, highlighting significant overlap and demonstrating the complexity of the task. This overlap confirms that straightforward methods like naive thresholding, as discussed by Meissen et al.[1], are insufficient, thereby underscoring the necessity for sophisticated generative synthesis techniques.

Supplementary Fig. 1: Pixel Intensity Distribution Comparison. This figure reveals a substantial overlap between the pixel intensity distributions of healthy (teal) and pathological (red) tissues, illustrating that our evaluation setup goes beyond mere intensity-based anomalies. The significant overlap suggests that simple thresholding methods would likely be ineffective for distinguishing between these two tissue types, emphasizing the need for more sophisticated diagnostic techniques.

R1.6: “I am a bit concerned about the generalization of the presented results beyond the brain. The brain itself is rather easy to interpret, there are well-defined homogenous regions, and rather regular anatomy where anatomical differences and diseases are recognised from local intensity changes. (See my comment before.) The authors simplify this problem by somehow pre-aligning the scans (I assume?) and only inspecting and evaluating a single well-defined slice of the whole 3D volume...”

A.1.6: We respectfully note that interpreting brain images is complex due to the subtlety of diseases and abnormalities, which can manifest as both local intensity changes and structural alterations like atrophy. This complexity is further confounded when analyzing T1-weighted images, where certain pathologies may not be as conspicuously visible as in FLAIR images (please refer to Supplementary Fig. 1). Furthermore, the brain’s plasticity means that the same functional areas can vary significantly in location across individuals, adding another layer of complexity to its interpretation. Nevertheless, while our experiments focus on brain MRI, the principles underlying our methodology are not specialized exclusively for this organ.

R1.7: “...I wonder: are the scores applicable to synthetically generated 3D data at all? And to more complex diagnostic tasks? (I believe they are?) Are the scores identify shortcomings of 2D generative algorithms that typically look nice in every single axial slice, but not so much when inspected under a 90 degree angle. I.e., how would a 3D version of these scores rank a stack of individually synthesised 2D images vs. a stack obtained from an algorithm generating 3D images? I understand that means for generating 3D data are still limited, but it would be nice to have at least a discussion of the 2D related shortcomings added to the paper...”

A.1.7: We thank the reviewer for raising an insightful point regarding the applicability of our evaluation metrics to synthetically generated 3D data. The reviewer is correct in noting that 2D generative algorithms, while potentially producing visually appealing results in individual axial slices, may not maintain spatial integrity when these slices are inspected from different angles, such as coronal or sagittal views [2]. Our proposed metrics are indeed applicable to 3D data and can be computed across the three different views—axial, coronal, and sagittal. This approach will highlight whether 2D methods that perform well in one specific view might fail to align slices into a coherent 3D representation of an organ. Such an analysis will provide a clearer indication of the shortcomings of 2D generative methods in maintaining 3D spatial coherence. We agree that the computational means for generating 3D data are still somewhat limited, which poses challenges for robust 3D synthesis. However, incorporating a discussion of these limitations and exploring potential evaluation strategies for 3D data will certainly enrich the paper. We added this extension to the discussion (lines 486-492 in the revised manuscript): *“Additionally, while 2D generative models produce high-quality slices, they often fail to maintain spatial integrity in a full 3D context [49]. Evaluating these models across axial, coronal, and sagittal planes helps assess their coherence in constructing accurate 3D structures, enhancing the utility of AI in complex diagnostic tasks.”*

R1.8: *“...Similarly, it would be good to have discussion of application domain related shortcomings, and to learn from the authors about the potential for a generalization to other situations, like the anomaly detection in abdominal CT. Right now this is a paper about anomaly detection in neuroimage data. (BTW: the title does refer to the limitations that is “image data” at all, which even exaggerates to the contrast to the brain image anomaly experiments.)”*

A.1.8: As detailed in **A.1.7**, the proposed metrics are not specifically designed for neuroimaging and can therefore be expected to also be applicable to other organs and imaging modalities, such as abdominal CT. Regarding the title of the paper, we recognize the need for greater specificity to accurately reflect the focus on brain MRI. To address this, we have adjusted the title to better match the scope of the experiments detailed in this study: *“Evaluating Normative Representation Learning in Generative AI for Robust Anomaly Detection in Brain Imaging”*.

R1.9: *“I would agree with all that is said in the intro. But is it worth to introduce (to define?) what normative learning is in the intro as well?”*

A.1.9: We have now incorporated a brief explanation of normative learning in the introduction and Fig. 1 to enhance clarity and understanding for readers: *“[...] in normative representation learning, a concept driven by data to uncover concise characteristics of a healthy population.”* Please also refer to **A.1.2**.

R1.10: *“p2 There is quite some prominence for the Generative AI algorithms (which is all good). But, they are no more than means for introducing and justifying the novel scores introduced here. Can you introduce the score you propose at the same level as the generative AI algorithms? Offering some high level insights or intuition? At present, the metrics are introduced in a part of a paragraph in the results section.”*

A.1.10: Thank you for your constructive feedback. We have revised the manuscript to incorporate a new section that presents our proposed metrics in a high-level, abstract manner. In summary, the RQI measures the perceived image quality, the AHI measures the distance of the pseudo-healthy restorations to the healthy distribution, and CACI measures the effectiveness in maintaining fidelity in healthy regions while correcting pathological areas, as detailed in **A.1.2** and lines 69-87 of the revised manuscript.

R1.11: *“The experiments are exclusively about the brain, a rather well structure organ, and mostly about local intensity. Does any of this apply to topological chances as well? (Think about vascular structures: bifurcation vs. trifurcation.) What would these scores look like for an abdominal CT?”*

A.1.11: Thank you for raising this insightful point. In our study, we aimed to cover a wide range of anomaly appearances to mitigate the risk of evaluating only hyper-intensities in brain images. Our analysis indeed encompasses not only local intensity anomalies, but also structural changes, such as atrophy (notably enlarged ventricles in the FastMRI+ dataset), which are purely structural and do not introduce intensity outliers. Additionally, our methods correct for shifts such as mass effect caused by tumors or secondary *vacuo* atrophy observed, for example, after ischemic stroke. Due to space constraints in the main manuscript, we were unable to include a detailed discussion of all anomaly classes we analyzed. We have attached the complete table encompassing the results on all anomaly types in Supplementary Table 1, along with visual results illustrating the structural corrections the methods facilitated in Supplementary Fig. 2. We believe these additional materials will provide a comprehensive view of the capabilities and applicability of our scoring metrics across a variety of contexts, including potentially in abdominal CT scenarios.

Supplementary Table 1: Anomaly Detection Performance on FastMRI+. The models are assessed based on the number of detections out of the total number of samples ($/N$) and F1 scores for various pathologies: absent septum pellucidum (ASP), craniotomy (Cran.), dural thickening (DT), edema, encephalomalacia (Enc.), enlarged ventricles (EV) intraventricular substance (IvS), lesions (Les.), posttreatment changes (Post.), resections (Res.), sinus opacification (Sinus), white matter lesions (WML) and mass. Best results are shown in **bold**.

Method	ASP		Cran.		DT		Edema		Enc.		EV		IvS		Les.		Post.		Res.		Sinus		WML		Mass			
	/1	F1 \uparrow	/15	F1 \uparrow	/7	F1 \uparrow	/18	F1 \uparrow	/1	F1 \uparrow	/19	F1 \uparrow	/1	F1 \uparrow	/22	F1 \uparrow	/44	F1 \uparrow	/10	F1 \uparrow	/2	F1 \uparrow	/5	F1 \uparrow	/26	F1 \uparrow		
AE	0	0.00	5	6.26	2	4.76	0	0.00	0	0.00	0	0.00	0	0.00	1	2.27	14	3.81	1	0.95	2	8.76	0	0.00	0	0.00	0	0.00
VAE	0	0.00	12	14.66	4	20.83	2	4.07	0	0.00	7	11.81	1	15.38	9	4.90	29	14.96	8	16.13	2	16.67	0	0.00	16	13.37	0	0.00
LTM	0	0.00	13	18.75	5	29.75	4	11.48	1	22.22	16	44.75	1	15.38	10	6.07	30	11.76	8	16.87	2	14.22	1	1.02	19	16.97	0	0.00
f-AnoGAN	0	0.00	14	19.19	3	9.54	2	3.44	0	0.00	13	22.82	1	12.50	8	3.68	30	12.08	8	17.78	2	9.16	2	1.58	16	12.36	0	0.00
SI-VAE	0	0.00	11	14.47	4	19.31	0	0.00	0	0.00	9	15.70	1	18.18	6	3.97	27	9.44	8	24.55	2	11.42	2	6.03	12	7.08	0	0.00
RA	1	15.38	13	34.78	6	52.65	12	45.56	1	66.67	18	77.54	1	50.00	17	29.50	35	30.78	10	54.32	2	26.67	5	15.50	21	30.78	0	0.00
DDPM-G	0	0.00	14	16.86	7	47.02	5	9.07	1	28.57	12	22.70	1	13.33	11	5.32	34	14.32	8	17.33	2	11.26	2	2.87	17	12.01	0	0.00
DDPM-S	1	14.29	9	14.04	6	38.48	14	35.51	1	40.00	17	51.23	1	28.57	16	16.92	32	15.94	10	33.21	1	1.72	3	8.14	22	12.42	0	0.00
ceVAE	0	0.00	14	16.99	4	22.70	3	4.52	0	0.00	4	5.91	1	20.00	10	5.05	32	14.76	7	15.00	2	15.88	0	0.00	19	14.12	0	0.00
MorphAEus	0	0.00	13	17.10	6	37.85	9	17.38	1	40.00	10	15.77	1	28.57	15	10.94	35	15.51	10	23.47	2	11.54	2	3.02	22	17.24	0	0.00
MAE	0	0.00	12	18.48	3	8.81	2	4.63	0	0.00	7	17.37	1	15.38	7	3.71	29	14.55	7	24.23	2	15.11	0	0.00	14	13.66	0	0.00
pDDPM	1	16.67	12	20.39	7	50.87	17	49.46	1	66.67	13	27.56	1	33.33	19	21.22	40	18.67	10	37.86	2	5.35	5	17.70	25	29.11	0	0.00
PHANES	1	18.18	14	36.15	7	51.31	16	43.28	1	100.00	19	80.51	1	40.00	21	27.31	40	29.65	10	49.33	1	10.00	2	4.07	24	33.89	0	0.00
autoDDPM	1	25.00	13	37.03	6	50.27	16	45.78	1	66.67	18	40.84	1	66.67	21	36.30	40	38.97	10	49.44	1	14.29	5	22.16	26	49.57	0	0.00

Supplementary Fig. 2: We show the application of three different anomaly detection techniques on MRI brain scans. Our evaluation goes beyond mere intensity-based anomalies. These models excel in identifying and visualizing a variety of structural anomalies. This includes atrophy, such as enlarged ventricles, changes following ischemic strokes, and mass effects due to tumors.

**R1.12:** “Fig 2, and elsewhere: Given that your anomaly detection heavily relies on intensity differences -
can you add a intensity outlyingness estimation to your comparisons? In your example of brain anatomy you
would get rather far with classical generative probabilistic models that model anatomy with atlas templates
and estimate local tissue-specific outlyingness by calculating tissue-specific intensity distributions. Can you
compare? (I don’t ask you to compare against this specific method, but just to give you a pointer to the body
of literature I refer to: <https://pubmed.ncbi.nlm.nih.gov/11513020/>.”

**A.1.12:** As previously mentioned in **A.1.5**, our study primarily explores the capabilities of generative
AI models for synthesizing anatomical images, not direct anomaly detection. We recognize the value of
classical generative probabilistic models for brain anatomy, as referenced, but they are beyond the scope of
our manuscript. For a detailed response please refer to **A.1.5** and Supplementary Fig. 1 where we show
the pixel intensity distributions of health and pathology are highly overlapping, which makes the detection
of anomalies difficult. Please also refer to **A.1.11**, Supplementary Table 1, and Supplementary Fig. 2
where we show the results on many different structural and intensity-based anomalies in brain MRIs.

**R1.13:** “p 4, l 143: I understand the details are in the method section, but can you mention the numbers
of the cases (approximative) and the number of different test cases?”

**A.1.13:** We have now added the numbers of cases used for training and testing in the mentioned
Section (lines 170-179 of the revised manuscript): “In our evaluation of generative AI models, we uti-
lized normal T1-weighted brain MRI datasets, FastMRI+ with 176 scans and 581 samples from IXI
(<https://brain-development.org/ixi-dataset/>), for model training. For the evaluation phase, we focused
on two key datasets: the enhanced FastMRI+ dataset, which encompasses a wide spectrum of 171
brain pathologies, and 420 subjects from the ATLAS v2.0 dataset, known for its diverse range of stroke
lesions.” To ensure transparency and facilitate reproducibility, we have now published the code here:

<https://github.com/compai-lab/2024-ncomms-bercea.git>, which includes all details concerning the archi-
tectures, hyper-parameters, training procedures, and the exact cases used for training and evaluation.

**R1.14:** “Fig 2: I am not sure this figure is a good summary. What is the brain visualized for?”

**A.1.14:** We think the reviewer meant Figure 1. We have redesigned Figure 1 to better align with conven-
tional scientific standards.

**R1.15:** “p 5, 172: You describe your three metrics and mention that they need to be holistically - and
qualitatively - interpreted to provide a comprehensive overview. As this is about metrics and expressing
diffuse statements about quality into simple and quantitative representations I wonder: can you offer some
summary statistic that merge the three scores? and p 6, l 238: what is “balanced” normative learning? You
are also mentioning the integration of the three scores three lines earlier. Is this a plain numeric summation
you are recommending?”

**A.1.15:** In this context, “balanced” normative learning refers to balancing the fidelity of the recon-
structions with the capacity to detect and replace anomalies, which are typically conflicting goals in the
literature [3]. We have now proposed a way to merge the metrics in the revised manuscript (please refer to
**A.1.4** and lines 233-238 and 617-624 in the revised manuscript. However, the analysis of individual scores
still provides unique insights into the models performance.

**R1.17:** “p 7, 305: “designed to evaluate the deep under306 standing of normal anatomy” => ?? rephrase”

**A.1.17:** We corrected to “designed to evaluate *how well AI models learn the underlying, normal anatomy.*”

**R1.18:** “p 8, 383: LPIPS is not defined. As a reader I don’t like to look up this reference 47. I understand
that your contribution is in the proposal to normalize this score, but can you offer those of your readers
who are unwilling to access the paper by Zhang et al a mathematical definition (or, in case this is difficult,
a brief summary of how this score is calculated)?”

**A.1.18:** Thank you for pointing this out. We have added a conceptual description of how LPIPS is
calculated in the Methods section (lines 556-566 in the revised manuscript):

“**Req. i. Restoration Quality Index (RQI)** evaluates restoration quality by semantically comparing
two images using the Learned Perceptual Image Patch Similarity (LPIPS). LPIPS is a metric that quantifies
perceptual similarity between images in a way that aligns closely with human judgment. This metric is
derived from the internal activations of deep convolutional networks, which, although trained on high-level
classification tasks, have shown to be effective in representing perceptual similarity [50].”.

To maintain the focus and readability of our manuscript, we further refer to reference [50] as it exten-
sively covers the foundational aspects of this metric.

**R1.19:** “p8, l 397: So this score requires having a segmentation of the lesion/anomaly? But if I have a fully
annotated test set, then I could go for an evaluation of false positives or negatives and even segmentations
scores, couldn’t I? Maybe it would help if you would explicitly describe (formalize?) how you foresee that
your scores are employed.”

**A.1.19:** Only the CACI score requires segmentation maps, while both the RQI and AHI can be calculated
from unlabeled images. These metrics are uniquely suited to holistically assess a model’s ability to **learn**
**normal anatomy**, while evaluating false positives or negatives with a fully annotated test set is limited
to the assessment of anomaly detection capabilities. We have now further improved the descriptions of our
scores in the Methods section for clarity. Please refer to **A.1.3** and lines 506-624 in the revised manuscript.

**R1.20:** “p 8, l 404. So you have about 700 datasets (minus the 235 that you comment on on the next
page). You only use one 2D slice for the extraction. Do you do any preprocessing like alignment to restrict
the variability of the appearance of these 700 slides? A rigid or affine registration of the 3D volumes? Or a
nonlinear one ? (I hope not!) Also, please report physical dimensions - pixels are about 1 mm² in size?”

**A.1.20:** Our preprocessing is minimal, including padding, resizing to 128 × 128 pixels, and intensity
normalization using the 98th percentile to a [0,1] range. We intentionally preserve the inherent variability in
the data, following the original datasets. For the *Atlas* dataset, images have a resolution of at least 1mm³,
pre-aligned to the MNI-152 template, and defaced. The *fastMRI+* dataset is 2D of poor axial resolution, also
defaced and used in its original form due to the extensive interpolation required for alignment to standard
templates (e.g., to isotropic 1mm³). We have revised the data-processing paragraph in the revised Methods
section (lines 653-665 in the revised manuscript) to reflect this description.

Reviewer 2:

R2.1: “Strengths: - This study addresses and interesting and a significant topic of anomaly detection with generative AI methods from the evaluation perspective with new metric. - Normal T1W brain images were used for experiments (IXI and fastMRI datasets), and for evaluation enhanced FastMRI and ATLAS v2.0. This strategy allows authors to test the capabilities of different models in detecting and localizing different anomalies. - The authors present three key metrics: RQI, AHI, CACI. These metrics offer some nice perspectives for the generative models.”

A.2.1: Thank you for your positive feedback. We especially appreciate your excellent suggestion and, as a result, conducted a comprehensive, multi-reader randomized study with 16 neuroradiologists to compare our proposed metrics against expert evaluations. This now forms a substantial new contribution to this manuscript.

R2.2: “Weaknesses: [...] Figure 1 has some styles which are not understandable. It looks like a magazine cover, rather than a scientific paper figure.”

A.2.2: We have redesigned Figure 1 to better align with conventional scientific standards.

R2.3: “it seems these three metrics (proposed) should be evaluated together, rather than in isolation (as authors themselves approve), but there is no clue if one single metric can be derived from these three still satisfying important properties like between 0-1, and somehow probabilistic and etc.”

A.2.3: Each proposed metric assesses unique facets of learning the normal anatomy. However, they should be evaluated together to validate the performance of different methods, as we have originally shown in Figure 4. As suggested, we have further clarified and formalized the integrative approach to combine these metrics in the Sections “Normative Learning Evaluation” (lines 233-238) and “Methods” (lines 617-624) of the revised manuscript: “*To fuse the metrics, we propose a harmonic mean between RQI and CACI, averaged with AHI. This approach balances image quality and anomaly correction while mitigating the impact of near-zero AHI scores on the overall evaluation (see Equation 4.)*” and “**Integrating the RQI, AHI, And CACI offers a comprehensive evaluation of generative models in medical imaging. However, individual assessments of these metrics still provide unique insights into specific aspects of model performance, highlighting strengths and areas for improvement. For a balanced overall evaluation, we propose a combined metric calculated as:**”

$$RQI, AHI, CACI = \frac{2 \times \left(\frac{RQI \times CACI}{RQI + CACI} \right) + AHI}{2}. \quad (2)$$

All three metrics have values between 0 and 1. We included this information in the revised Methods section (lines 555-625 of the revised manuscript).

R2.4: “There are many unjustified sentences and words like profound etc...I do not think the study implies such extreme excellence. yet, maybe initial results with good metrics, but they do not show clinical implications yet.”

A.2.4: We carefully revised the claims of the papers and edited them where it was appropriate: “*Our analysis marks a significant shift in generative AI for medical imaging, advocating for ~~a~~ profound emphasis on normative representation learning.*” and “*Clinically, the implications are ~~profound~~ considerable. Models that master normative learning can discern subtle pathological nuances*”.

R2.5: “not sure how these metrics are affected from preprocessing steps? MRI has a lot of challenges to address before the actual processing.”

A.2.5: Indeed, metrics such as LPIPS or FID might be influenced to some extent by preprocessing steps. However, we have opted for a consistent pre-processing scheme for fair comparisons of different models.

R2.6: “potential biases coming from different networks when FID is used are not discussed.”

A.2.6: Thank you for raising this point. We acknowledge this concern and have discussed the multifaceted nature of the FID metric in our revised manuscript (lines 426-432): “*Minor inconsistencies in the AHI might stem from its use of the FID, which evaluates not only the “health” status of images but also other aspects such as sampling diversity and domain alignment between the evaluated sets*”.

R2.7: “AHI includes FID, and FID is not capturing semantic similarity, but feature similarity. Therefore, I am not sure if AHI is really capturing the semantic similarity. More elaboration are needed.”

A.2.7: We agree with the reviewer and confirm that the AHI measures statistical differences in feature distributions, which we use to evaluate how methods align transformed pathological sets with healthy distributions. For assessing semantic similarity, we have originally introduced the Reconstruction Quality

Index (RQI), which evaluates pairwise perceptual similarity between images. We have clarified the distinct
roles in the revised manuscript to ensure their purposes are understood (please also see A.1.2) and lines
69-87 of the revised manuscript.

**R2.8:** “*There seems to be no expert evaluation other than some elaboration on certain scores. Why not*
*run a Turing test with radiologists (multiple?)*”

**A.2.8:** Thank you for your excellent suggestion regarding the inclusion of expert evaluations. Based on your
feedback, we have now conducted a comprehensive study involving 16 radiologists (3 experienced, board-
certified radiologists and 13 residents) to assess the performance of our AI-generated images against real
images. This evaluation has become a significant component of our study and one of its main contributions.
Details of this analysis and its findings are now included in the revised manuscript and can be found in the
updated Section “Clinical Validation” (lines 361-436 of the revised manuscript). Key findings include:

- 1. Our proposed metrics, RQI and AHI, showed good correlation with radiologists’ assessments of image
realness, quality, and health status.
- 2. AI models closely mimicked real images, with models like AutoDDPM often misleading radiologists.
- 3. AI models closely reproduced healthy images, especially for methods like RA, achieving similar scores to
real images.
- 4. Scores for real images also displayed a range of variability, indicating that even annotated pathology-free
images did not consistently receive uniform assessments of being ‘healthy’.

Reviewer 3:

**R.3.1:** “*In this article, the authors present new metrics to evaluate deep generative models for unsupervised*
*anomaly detection (UAD) in medical images. These metrics assess how well deep generative models learn the*
*healthy distribution, as opposed to their ability to detect anomalies, as is done usually. The metrics are used*
*to evaluate about fifteen approaches in the context of brain lesion detection in structural MRI. Designing*
*new ways to evaluate UAD is crucial to understand how these approaches actually work and fully assess*
*their potential.*”

**A.3.1:** We appreciate the reviewer’s recognition of our contribution.

**R.3.2:** “*The paper is well written and convincing, but I would have appreciated a deeper analysis of the*
*metrics. In particular, it would be interesting to better know how they behave depending on the severity of the*
*lesions. These could be done pushing further the evaluation with the stroke dataset or using simulated data.*”

**A.3.2:** Following your suggestion, we have now broadened our original analysis of lesion sizes—small,
medium, and large—to include a comprehensive table that details performance across various disease types
(Supplementary Table 1), each distinguished by unique phenotypes and severity levels. Additionally, the
manuscript now includes detailed examples demonstrating how AI methods can effectively address complex
pathological changes, such as mass effects or atrophic shifts post-stroke (Supplementary Fig. 2). Please also
refer to our response in A.1.12.

To further substantiate the analysis of our proposed metrics and confirm the clinical relevance of our
findings, we incorporated results from a comprehensive study involving 16 radiologists. This assessment
focused on the realness, quality, and health status of images, highlighting that our proposed metrics
strongly correlate with clinical evaluations and demonstrating that AI methods can generate realistic
pseudo-healthy images. Details of this evaluation are provided in the “Clinical Validation” section (lines
361-436 of the revised manuscript). Please also refer to A.2.8.

**R.3.3:** “- l.35: ‘*The true innovation of Generative AI lies in normative learning, yet its evaluation often*
*skews towards anomaly detection, leading to biased or incomplete assessments.*’ *I would reformulate this*
*sentence as it gives the impression that evaluating an approach (normative learning) using the task it was*
*developed for (anomaly detection) is a bad thing. It is not, but it should not be the only evaluation criterion.*”

**A.3.3:** We rephrased to: “*However, the evaluation of generative AI methods ~~often tilts disproportionately~~*
*~~yet its evaluation often skews~~ towards anomaly detection, leading to biased or incomplete assessments.*
(lines 43-46 of the revised manuscript).”

**R.3.4:** “- l.39: ‘*certain approaches optimize solely for hyper-intense lesion detection [18]*’ *Could you be*
*more specific, i.e. which approach(es) from all the ones being compared in [18] are you targeting?*”

**A.3.4:** All approaches in [18] were subject to the same hyper-intense post-processing, please see section 3.3
in the referenced manuscript. The results without post-processing are shown in Table VIII in the appendix

of [18]. All methods without hyper-intense-specific post-processing achieve very poor performance.

R.3.5: “- l.41: ‘anchor their design principles around large hyper intense tumors [19]’ In this approach, training is also performed using healthy subjects only. Please explain why you think it is specific to a type of lesion.”

A.3.5: The approach in [19] enables the model to simulate anomalies through noise patterns, despite being trained solely on healthy images. This coarse Simplex noise replicates the scale and intensity variations typical of such anomalies in MRI scans. However, this specificity limits its effectiveness with other types of anomalies, leading to documented detection failures [4, 5]. We rephrased the paragraph in the introduction to: “anchor their design principles (e.g., specific noise types) around large hyper-intense tumors [19]. (l. 48)

R.3.6: “- l.42: ‘resort to approximating the unknown anomaly distribution through self- or weakly-supervised strategies [20, 21].’ Do you imply that this is a bad thing? If so, why?”

A.3.6: We did not wish to imply that using these strategies to approximate anomaly distributions is inherently negative. Rather, our concern is with ensuring that the limitations regarding the scope of detectable anomalies is clear. We rephrased to (l. 50-55) “or ~~resort to approximating~~ approximate the unknown anomaly distribution through self- or weak-supervision [20,21]. While such methodologies can be advantageous for detecting specific pathologies, they often fall short in broader anomaly detection contexts [16].”

R.3.7: “- Fig. 2. Not having the AE displayed is a bit confusing, especially as in Table 1 the letter A refers to AE.”

A.3.7: We omitted the baseline AE as it is not a generative method, instead focusing on its generative counterpart, the VAE. We have updated Table 1 to further improve the clarity.

R.3.8: “- l.85: ‘reconstruction quality, which is essential for precise anomaly detection.’ Some works seem to imply something different, e.g. Lambert et al., ISBI 2021. As it is a point that you raise later in the result, you could moderate it here and insist on it later.”

A.3.8: We thank the reviewer for the suggestion. While we want to maintain the readability in the methods description we added this lines to the Interplay between Normative Learning and Anomaly Detection section (l. 348-360): “This finding aligns with existing literature [18,47], like the work of Baur et al.[47], which notes that methods with dense latent spaces, often producing less sharp reconstructions, can surprisingly outperform more precise counterparts”

R.3.9: “- Fig. 3. It might be useful to specify that the normative learning metrics were obtained for the whole dataset and do not correspond to the exemplar image being displayed. It would be especially useful as a lesion is visible in the AutoDDPM reconstruction while the AHI is very high, while for example no lesion seems visible in the DDPM-G reconstruction but the AHI is low.”

A.3.9: We updated the caption of Figure 3 as follows: “**Normative Learning Evaluation.** The central pathological MRI image is encircled by the pseudo-healthy restorations of different models, with their respective dataset-wide metric scores (RQI, AHI, CACI) radiating outward. ~~The color intensity and height of the bars represent the distribution of the scores, with Segments with darker and higher values indicate better performance, This highlightsing their effectiveness in image restoration, anomaly normalization, and preservation of healthy tissue.~~”

R.3.10: “- l.154: Following-up on the previous comment, it would be good to clarify whether normative learning metrics are computed for a whole dataset or can be obtained per image.”

A.3.10: RQI and CACI are computed on a per-image basis, while the AHI metric evaluates distributional differences between sets, and is assessed using the entire dataset. To ensure a thorough evaluation of these methods, we recommend analyzing all metrics across a diverse range of images with various anomalies. We have outlined a high-level description of the metrics in the Introduction (lines 69-87 in the revised manuscript) and further clarified the mathematical definitions in the Methods section (lines 555-625). Please also refer to **A.1.2**.

R.3.11: “- Table 1: How were the ranks obtained, i.e., based on what criteria? This information is important as later in the text (l. 200–209) the ranks are commented upon, but if all the methods have very similar scores, then the rank may not mean much.”

A.3.11: The ranks were computed on the mean results on the anomaly detection performance. We changed the caption of Table 1 to: “[...] Ranks (wrt. to mean performance) are provided for both tasks.”

**R.3.12:** “- Fig. 4: It is not easy to see which line connects which metric/task. Also, one line, starting on
the left and going in a horizontal direction, seems to stop before reaching another point.”

**A.3.12:** Thank you for pointing this out. The lines at the border (left and right horizontal lines) follow
the same trajectory in both directions, which might make them appear as if they stop. To enhance clarity,
we have added bi-directional arrows on these lines.

**R.3.13:** “- l.277: ‘hinting at overfitting to specific scenarios rather than wide-ranging disease detection.’
How do you explain this behaviour, knowing that these approaches, as the others, are trained on healthy
data?”

**A.3.13:** Training solely on healthy data does not necessarily ensure that models accurately learn the
healthy distribution [6]. When methods are trained in an unsupervised manner without biases for specific
expected anomalies, they might still struggle to faithfully reconstruct healthy data, as evidenced by the
CACI metric. This leads to failures in detecting subtler or small pathologies, for which a high-resolution
synthesis is needed. This challenge is seen in all methods within the Average 4 Group in Figure 4, leading
to increased false positives due to reconstruction errors and failures in identifying subtle anomalies.

**R.3.14:** “- l.353: Almost nothing is said about the generative models being evaluated (2D/3D, architecture,
hyper-parameters, losses, training procedure, etc.). Even though these are existing models, a minimum should
be said, e.g. whether everything was kept the same as in the paper being cited; if modifications were made,
which ones.”

**A.3.14:** Thank you for your comment. We have strived to keep the implementations of the genera-
tive models as close as possible to their original descriptions in the literature. To ensure transparency
and facilitate reproducibility, we have now published the code here: [https://github.com/compai-lab/
2024-ncomms-bercea.git](https://github.com/compai-lab/2024-ncomms-bercea.git), which includes all details concerning the architectures, hyper-parameters, training
procedures, and the exact cases used for training and evaluation.

**R.3.15:** “ l.403: Similarly, little is said about the data. For example, no demographic information is
provided, the MRI sequence is barely mentioned, it is not clear whether training is done with all 2D slices
or just mid-axial slices.”

**A.3.15:** We have specified in the Results and Dataset Description sections that our analysis focused on
T1-weighted (T1w) MRI sequences. Additionally, in the Pre-processing section (lines 653-665), we clarified
that only mid-axial slices were used. For comprehensive demographic details and additional data charac-
teristics, we have included links to the public datasets, where complete information is readily available.

**R.3.16:** “l.431: How are the residual images processed to compute the DSC?”

**A.3.16:** To compute DSC, we first binarize the continuous residual maps and then compare them with the
ground truth. We follow common practice in the literature by maximizing the binarization threshold across
the dataset to report the maximum Dice value ($[Dice]$), avoiding reliance on a specific operating point.

References

[1] Meissen, F., Kaissis, G. & Rueckert, D. *Challenging current semi-supervised anomaly segmentation*
*methods for brain mri*, 63–74 (Springer, 2021).

[2] Durrer, A. *et al.* Denoising diffusion models for 3d healthy brain tissue inpainting. *arXiv preprint*
*arXiv:2403.14499* (2024).

[3] Bercea, C. I., Neumayr, M., Rueckert, D. & Schnabel, J. A. *Mask, stitch, and re-sample: Enhancing*
*robustness and generalizability in anomaly detection through automatic diffusion models* (2023).

[4] Bercea, C. I., Wiestler, B., Rueckert, D. & Schnabel, J. A. Reversing the abnormal: Pseudo-healthy gen-
erative networks for anomaly detection. *Medical Image Computing and Computer-Assisted Intervention*
(2023).

[5] Bercea, C. I., Wiestler, B., Rueckert, D. & Schnabel, J. A. Diffusion models with implicit guidance for
medical anomaly detection. *arXiv preprint arXiv:2403.08464* (2024).

[6] Bercea, C. I., Rueckert, D. & Schnabel, J. A. *What do aes learn? challenging common assumptions in*
*unsupervised anomaly detection*, 304–314 (Springer, 2023).

Evaluating Normative Representation Learning in Generative AI for Robust Anomaly Detection in Brain Imaging – Point-by-point response –

Cosmin I. Bercea^{1,2*}, Benedikt Wiestler^{1,3}, Daniel Rueckert^{1,3,4}, Julia A. Schnabel^{1,2,5}

¹Technical University of Munich, Munich, Germany.

²Helmholtz AI and Helmholtz Center Munich, Munich, Germany.

³Dept. of Neuroradiology, Klinikum Rechts der Isar, Munich, Germany.

⁴Imperial College London, London, UK.

⁵King's College London, London, UK.

*Corresponding author(s). E-mail(s): cosmin.bercea@tum.de;

We thank the reviewers and editor for their time and constructive feedback, which has significantly improved the quality of our work. In response, we have carefully addressed the editorial requests (outlined in the Author Checklist) as well as the reviewers' comments (highlighted in the manuscript, with **removed** and **added** text indicated in color).

Reviewer 1:

R1.1: *"Thank you for the responses. They helped me in clarifying about the paper. This is a very nice study that can be impactful in our community!"*

A.1.1: We thank Reviewer 1 for their time and feedback which helped to improve our manuscript.

Reviewer 2:

R2.1: *"Adding several radiologists to do user study is great, the paper becomes very solid, and field will get benefit. Thanks for adding this component. Some further minor adjustments are necessary: – supplementary table 1, all the references are missing – figures and tables are nice, are they reproducible from code ? or powerpoint/paint ? "*

A.2.1: We thank Reviewer 2 again for their time and insightful suggestions that helped improve our manuscript. We fixed Supplementary Table 1 to include the missing references. We have provided scripts on our GitHub to generate the more complex plots.

Reviewer 3:

R.3.1: *" l.55 'This approach marks a shift in the assessment of generative AI, positioning it not just as a tool for disease identification but as a system for understanding the spectrum of human health in medical imaging.' The last part of the sentence appears like an overstatement not supported by results, please rephrase."*

A.3.1: We rephrased to: "This approach marks a shift in the assessment of generative AI, positioning it not just as a tool for disease identification but as a system for **understanding the spectrum of human health evaluating the realness and plausibility of generated counterfactuals** in medical imaging."

R.3.2: “Several acronyms are not defined, e.g. MAEs (l.120), LTM (l.154)”

A.3.2: We apologize for this omission. We have now introduced the acronyms.

R.3.3: “l.173 ‘To fuse the metrics, we propose a harmonic mean between RQI and CACI, averaged with AHI.’ This is a good thing to propose a way to combine the three metrics proposed but is this composite metric evaluated in the present work or some results presented? It does not seem to be the case. Please clarify and/or provide some results obtained with this composite metric to justify its soundness.”

A.3.3: We used the composite metric (noted as “RQI, AHI, CACI”) in Figure 4 (chord diagram and heatmap) to measure the interplay between normative representation learning and anomaly detection performance. We hope that this clarifies your question.

R.3.4: “l.234 ‘However, the most comprehensive insights are obtained when these metrics are collectively analyzed, with combinations such as CACI and RQI, and particularly the integration of all three metrics (RQI, AHI, CACI), demonstrating enhanced predictive anomaly detection power.’ Is the composite metric used in this analysis?”

A.3.4: Yes, this is the composite metric as explained in the Section Interplay Normative Learning and Anomaly Detection and Figure 4.

R.3.5: “Fig. 4: Please clarify the metric used to assess the ‘detection performance’.”

A.3.5: For the normative learning we used the metrics indicated on the Y axis (RQI, AHI, CACI) and for the anomaly detection we used the [Dice] score where pixel-level annotations were provided (Atlas) and the F1 score for the detection tasks on FastMRI+. We rephrased line 225 for clarity as follows:

“Our in-depth analysis, detailed in Figure 4, investigates the relationship between normative learning metrics and diverse anomaly detection tasks metrics ([Dice] or F1 as in Table 1), revealing key insights:”

R.3.6: “l.299 ‘To evaluate the performance of different AI algorithms and validate our quantitative metrics—RQI and AHI—we conducted a comprehensive test involving 16 radiologists.’ The authors could explain here why the CACI is not part of the evaluation (instead of having to wait until the end of the section).”

A.3.6: We moved the mentioned paragraph at the beginning of the section as suggested.

R.3.7: “Fig. 5 (a): What does the width of the violin plot correspond to? Please comment on the appearance of the plots for the MAE.”

A.3.7: The width of each violin plot represents the density of scores at different rating levels, with wider sections indicating a higher concentration of scores at those values. For the MAE model, the plots appear thinner due to the low variability in “Quality” scores (1.02 ± 0.14), reflecting consistent ratings. For “Realness” and “Health,” larger standard deviations suggest some divergence in opinions, likely clustering into two very distinct groups (mostly around 1 and a few at 4-5). We used Seaborn to generate these plots, which automatically normalizes the width based on density. We will release the anonymized poll results and the script used to generate the plots for transparency.

R.3.8: “l.420 ‘Additionally, while 2D generative models produce high-quality slices, they often fail to maintain spatial integrity in a full 3D context [49]. Evaluating these models across axial, coronal, and sagittal planes helps assess their coherence in constructing accurate 3D structures, enhancing the utility of AI in complex diagnostic tasks.’ This is a very generic statement that seems disconnected from the present work. Please rephrase to clearly present it as a limitation or future work as this analysis is not performed here.”

A.3.8: We rephrased as:

“Additionally, while 2D generative models produce high-quality slices, they often fail to maintain spatial integrity in a full 3D context [49]. Future work could evaluate these models across axial, coronal, and sagittal planes to assess their coherence in constructing accurate 3D structures, enhancing the utility of AI in complex diagnostic tasks.”

R.3.9: “l.497 Several data sets are mentioned when defining the metrics (D_{UH}, D_P, D_H). It would be good to specify what data from the IXI, FastMRI+ and ATLAS datasets are used to create them as it is currently unclear.”

A.3.9: We have now further specified the datasets in the Methods section as follows:

Healthy Data for Training: Two public T1w brain MRI datasets from healthy individuals D_H were employed for model training: IXI (<https://brain-development.org/ixi-dataset/>) (581 training samples) and FastMRI+[45] (131 training, 15 validation, 30 test samples). We kept 30 samples from the FastMRI+ as unseen healthy test samples D_{UH} .

Pathology Data for Evaluation: We used two public datasets containing several disease classes as our datasets D_P containing pathology: [...]

Please note that we also provided the exact cases used in the released CSV files with our framework.

R.3.10: “1.541 I insist that more should be written about the datasets (e.g., a summary table in the supplementary). It is commendable to release code, but it is not the job of the reader to check code, articles and websites to see whether there are large discrepancies in population demographics or image characteristics (field strength, sequence (MPRAGE? SPGR?), etc.)”

A.3.10: We have now added this additional information, summarized in Supplementary Table 2 in the Supplementary Materials:

Supplementary Table 2: Comparison of Brain MRI Datasets: Demographics, Imaging Characteristics, Disease Types, Annotations, Preprocessing, and Acquisition Types.

Characteristic	IXI	fastMRI+	ATLAS
Number of Subjects	581	5847	1271
Number of Annotated Scans	581	1001	655
Age Range (years)	20–86	Not specified	Not shared
Sex Distribution	Male = 277, Female = 342	Not specified	Not shared
Scanner Vendors	Philips, GE, Siemens	11 different scanners	Not specified
Scanning Location	3 hospitals in London, UK	5 clinical locations	Not specified
Field Strength (T)	1.5, 3.0	1.5, 3.0	1.5, 3.0
Image Sequences	T1 , T2, PD, MRA, DWI	Axial T1 , Axial T2, Axial FLAIR	T1
Disease Type	Healthy volunteers	Various pathologies	Ischemic stroke lesions
Annotation Type	None	Bounding box annotations	Pixel-wise segmentation Intensity standardization
Preprocessing	Not specified	Cropped for de-identification	Linear registration to MNI 152 Defacing
Acquisition Type	3D	2D Axial acquisition	3D

R.3.11: “1.552 ‘We used the annotations provided as bounding boxes by medical experts to assess the detection (#’det’) and precision (F1 score) of the models [16].’ This procedure should be described in greater details here (in the supplementary).”

A.3.11: We have now detailed the computation of the metrics in the main manuscript as follows:

We used the annotations provided as bounding boxes by medical experts to assess the detection (’#det’) and precision (F1 score) of the models [16]. We considered a detection to be a true positive (TP) if at least 10% of the pixels within the annotated bounding box were flagged as anomalous. False positives (FP) were calculated as the ratio of misdetected pixels on healthy tissue relative to correctly detected pixels within the anomaly box. Finally, we report the F1 score as: $F1 = \frac{1}{N} \sum_{i=0}^N \frac{2 \times P \times TP}{P + TP}$, where $P = \frac{TP}{TP + FP}$ and N is the number of test cases.

R.3.12: “1.565 When mentioning ‘the largest Dice coefficient’ please provide more details and/or add a reference.”

A.3.12: We have now added more details on the largest Dice coefficient as follows:

We stratified the test sets into small (first 25th percentile, < 71 pixels), medium, and large lesions (top 25th percentile, ≥ 570 pixels) for performance evaluation, with the largest Dice coefficient ‘[Dice]’ as the metric. [Dice] represents the theoretical maximum segmentation accuracy, achieved by a greedy search for the best residual threshold on the test set [8].

R.3.13: “1.569 The pre-processing performed, even if run by the dataset providers, should be explained in the supplementary. Again, this is not the readers’ job to fish for this information within multiple sources.”

A.3.13: We have now added this information to the new Supplementary Table 2.